# Bi-level optimization of shared manufacturing service composition based on improved NSGA-II

**Ying Wang, Peng Liu** *

School of Management, Shenyang University of Technology, Shenyang, China

* liupeng@sut.edu.cn

**Data Availability Statement:** All relevant data are within the manuscript and its Supporting Information files. The authors declare no conflict of interest. We declare that there is no conflict of interests regarding the publication of this article.

## Abstract

To address the issue of insufficiently comprehensive representation of service composition indexes in the shared manufacturing environment, service reliability, confidence, and other indexes are decomposed in detail to establish a composition evaluation system, and a shared manufacturing service composition optimization model based on bi-level programming is proposed. The model takes Quality of Service (QoS) as the upper objective function and service reliability, service confidence and task fit as the lower objective function. The upper objective function needs to be minimized, while the lower objective function needs to be maximized. To achieve the optimal composition scheme of shared manufacturing services, the Criteria Importance Though Intercrieria Correlation (CRITIC) is used to determine the weights of the indicators, and the improved Fast Elitist Non-Dominated Sorting Genetic Algorithm (Improved NSGA-II) is employed to solve the multi-objective optimization problem. Finally, the improved NSGA-II has a 23.33% increase in convergence speed and a 69.99% gain in operational efficiency when compared to the traditional NSGA-II. The viability and effectiveness of the improved NSGA-II have been demonstrated.

## 1. Introduction

With the introduction of a new generation of Internet technology, modern information technology, and modern management style, the manufacturing industry has developed into an advanced manufacturing industry with high knowledge and technology. With the arrival of the Industry 4.0 era, a new socialized manufacturing model based on the sharing economy has arisen: shared manufacturing. With the qualities of intensification, high efficiency, and flexibility, shared manufacturing is a new breed of industrial organization that applies the sharing concept to all facets of manufacturing. It does this by integrating dispersed and idle production resources, matching them flexibly, and dynamically sharing them with the demand side [1]. With the help of P2P collaboration, shared manufacturing can be used to expand the extent and scope of resource sharing, thereby improving the competitiveness of the manufacturing industry. Due to the increasing manufacturing demand and satisfaction of consumers, the manufacturing model has evolved from mass production to mass customization and personalization to meet increasingly diverse customer needs [2,3], and shared manufacturing is moving

**Funding:** This work was supported by the Research Project on Economic and Social Development of Liaoning Province under Grant 2024lslybkt-058, Liaoning Graduate Education and Teaching Reform Research Project under Grant LNYJG2022062. The Key Program of Social Science Planning Foundation of Liaoning Province under Grant L21AGL017.

**Competing interests:** The authors have declared that no competing interests exist.

in this direction. As a core part of shared manufacturing to provide services to consumers with personalized needs, a shared manufacturing service composition aims to go through a process of service composition and optimal selection, providing consumers with services to meet their personalized manufacturing needs [4]. The shared manufacturing platform divides production tasks into several sub-tasks, which are released on the demand side of shared manufacturing services [5]. In the sub-tasks matching stage, several candidate manufacturing resource services might finish the sub-tasks. The shared manufacturing platform needs to think about how to accurately and efficiently perform the task decomposition and composition to choose an ideal composition to finish the manufacturing activities.

The contributions of this paper are as follows: ① This paper establishes a bi-level programming model, which is different from previous bi-level programming models. The model targets the supply and demand sides of shared manufacturing, and the lower level indicators include service reliability, service credibility, and task coordination. The service reliability and credibility indicators are decomposed in detail, and the model reflects broader interests. ② This paper uses the Criteria Importance Though Intercrieria Correlation (CRITIC method) to determine the weights of indicators and make the lower indicators more fair and objective. ③ This paper has enhanced the genetic algorithm's selection, crossover, and mutation processes, as well as its efficiency and rate of convergence.

The remainder of this paper is organized as follows: Section 2 reviews related work. In Section 3, we introduce the problem description and construct the bi-level programming model. We propose the solution of the model in Section 4. A numerical example is given in Section 5. Section 6 gives some concluding remarks.

## 2. Literature review

### 2.1. Service composition

Service composition is the main topic of the current research concern. SHIRVANI M H el. [6] presented an iterative mathematical decision model for organizations to evaluate whether to invest in establishing information technology (IT) infrastructure on-premises or outsourcing IT services on a multicloud environment. In order to reduce overall costs and increase security in multi-cloud environment, SHIRVANI M H [7] formulated the web service composition problem as a bi-objective optimization problem with service cost and multi cloud risk perspectives in the ever-increasing multi cloud environment (MCE). TABALVANDANI M A N et al. [8] proposed a multi-objective particle swarm optimization model in multi-cloud scenarios to increase system reliability. Xie et al. [9] created a hybrid multi-objective optimization algorithm to solve the service composition optimization problems of production factor resources in an industrial internet setting. To improve time, cost, reliability, availability, and reputation, Liu et al. [10] suggested an evolutionary algorithm based on adaptive selection and inverse learning strategies. Peng et al. [11] created a CMSSS multi-objective optimization model to reduce the use of industrial energy while maintaining quality of service (QoS). Wang et al. [12] developed a new multi-objective optimization problem for the integration of urgent task-aware cloud manufacturing services based on service quality. Song et al. [13] took manufacturing service uncertainty into account and built a cloud-edge collaboration-based optimization framework for cloud manufacturing service composition. Some researchers also enhanced the method to create more efficient compositions. Que et al. [14] suggested a new immune genetic method based on information entropy. Liu et al. [15] suggested a deep deterministic policy gradient-based service composition approach to cloud manufacturing services.

## 2.2. Evaluation indicators of the service composition

To address the issues of service composition, it is first necessary to consider the indicators for evaluating service composition. Some scholars have established relevant multi-objective combination models from different perspectives [16–18]. In addition to studying fundamental attribute indicators such as time and cost, some scholars have also established models that include indicators such as service credibility and reliability [19–22]. Viriyasitavat et al. [23] advocated blockchain technology to transfer and validate the trustworthiness of services and partners. They created a BPM framework to show how BCT may be used to provide quick, dependable, and cost-effective service quality assessment in service portfolios. Guo et al. [24] built a multi-objective optimization model for cloud manufacturing service portfolios that took into account the interests of multiple subjects, using the demands of service demanders, platform operators, and service providers as constraints. Ren ML et al. [25] presented a synergy-based service composition approach in order to improve the social collaboration features of manufacturing services.

## 2.3. Non-dominated sorting genetic algorithm

To solve multi-objective optimization problems in a shared manufacturing environment, it is usually necessary to transform the multi-objective optimization function into a single objective optimization problem and solve it [26–28]. This type of method has the following problems: (1) There is usually a certain degree of contradiction and constraint relationship between their respective objectives, and the objective function is only the linear weighted sum of various indicators, which cannot obtain the corresponding equilibrium solution; (2) The optimization progress of each objective during the solving process is unpredictable and difficult to adjust. Therefore, domestic and foreign scholars have proposed non-dominated sorting genetic algorithms.

NSGA-II (Non-Dominated Sorting Genetic Algorithm) is an advanced evolutionary algorithm that specifically designed to solve multi-objective optimization problems [29]. Non-dominated sorting categorizes individuals in a group based on their performance on multiple goals, with priority given to retaining those who perform well on each goal. The Pareto solution set is made up of all combined non-dominated solutions. Using the Pareto solution set, decision makers can select the best option based on their preferences and real needs. To increase the NSGA-II algorithm's efficiency and produce a Pareto solution set that is more ideal, some researchers have made improvements to it [30,31]. This paper has improved the NSGA-II selection, crossover, and mutation processes by including some of the ideas from this, which has increased the algorithm's efficiency and convergence.

# 3. Shared manufacturing service composition modeling based on bi-level programming

## 3.1. Service composition problem description

Service composition in a shared manufacturing environment refers to the matching of optimal compositions of manufacturing resources and services for different manufacturing tasks on a shared manufacturing platform under multiple constraints and multi-objectives to fulfill the shared manufacturing tasks issued by the demand side.

The whole process of a shared manufacturing service composition is shown in Fig 1. First, we must decompose the complex manufacturing task on the demand side into multiple manufacturing sub-tasks $SMS_i$, based on the functional similarity of the task and the matching of the resources, i.e., the total task $SMS = \{SMS_1, SMS_2, SMS_3 \ldots SMS_i \ldots SMS_n\}$. Each $SMS_i$ has a corresponding set of candidate services $MMR_{i,j}, MMR_{i,j}$ means that the jth manufacturing resource completes the ith sub-task, in which each candidate service can complete the sub-

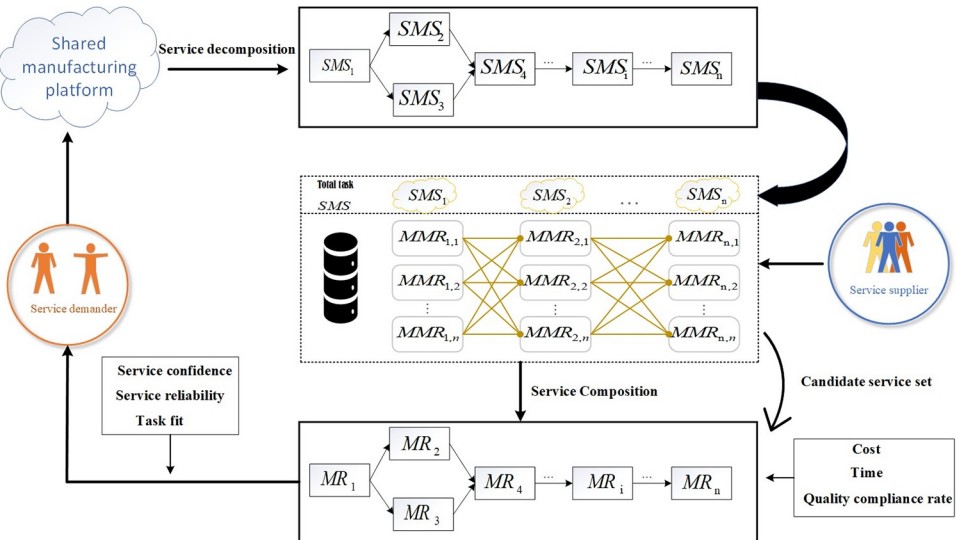

**Fig 1. Shared manufacturing services composition flowchart.**

task, and all the sub-tasks corresponding to the candidate services constitute a candidate service composition. Because of various factors, the time, cost, and other aspects consumed by each candidate service may differ when completing the same sub-task, so the time, cost, and quality pass rate of different service combinations to finally complete the manufacturing task are not the same, so in the face of complex manufacturing tasks, the optimal service combination can be chosen to maximize the interests of both supply and demand.

## 3.2. Evaluation indicators

Fig 2 displays the evaluation index system of the bi-level programming model proposed in this paper, which takes into account the demand and supply of the shared manufacturing service composition process as well as manufacturing resource characteristics. The index system is divided into two layers, with the upper-level indexes concentrating on the supply side of shared manufacturing services, such as cost, time, and quality qualification rate indexes. Lower-level indexes concentrate on the demand side of shared manufacturing services, such as service reliability, confidence, and task fit.

All of the above metrics are quantifiable values that may be acquired at the shared manufacturing platform to determine the multi-objective function and the accompanying constraints, and further descriptions of each metric are provided in Table 1.

## 3.3. Upper-level optimization metrics

**(1) Cost**

Suppliers incur certain expenses while offering services. The cost of shared manufacturing services primarily involves the cost of manufacturing resource implementation $C_{ir}(R_{ij})$ and logistics and transportation $C_{xi}(R_{ij}, R_{(i+1)k})$.

The formula is as follows:

$$C = C_{ir} + C_{xi} = \sum_{i=1}^{m}\sum_{j=1}^{n_i} x_{ij}C_{ir}(R_{ij}) + \sum_{i=1}^{m}\sum_{j=1}^{n_i}\sum_{k=1}^{n_{i+1}} x_{ij,(i+1)k}C_{xi}(R_{ij}, R_{(i+1)k}) \tag{1}$$

**(2) Time**

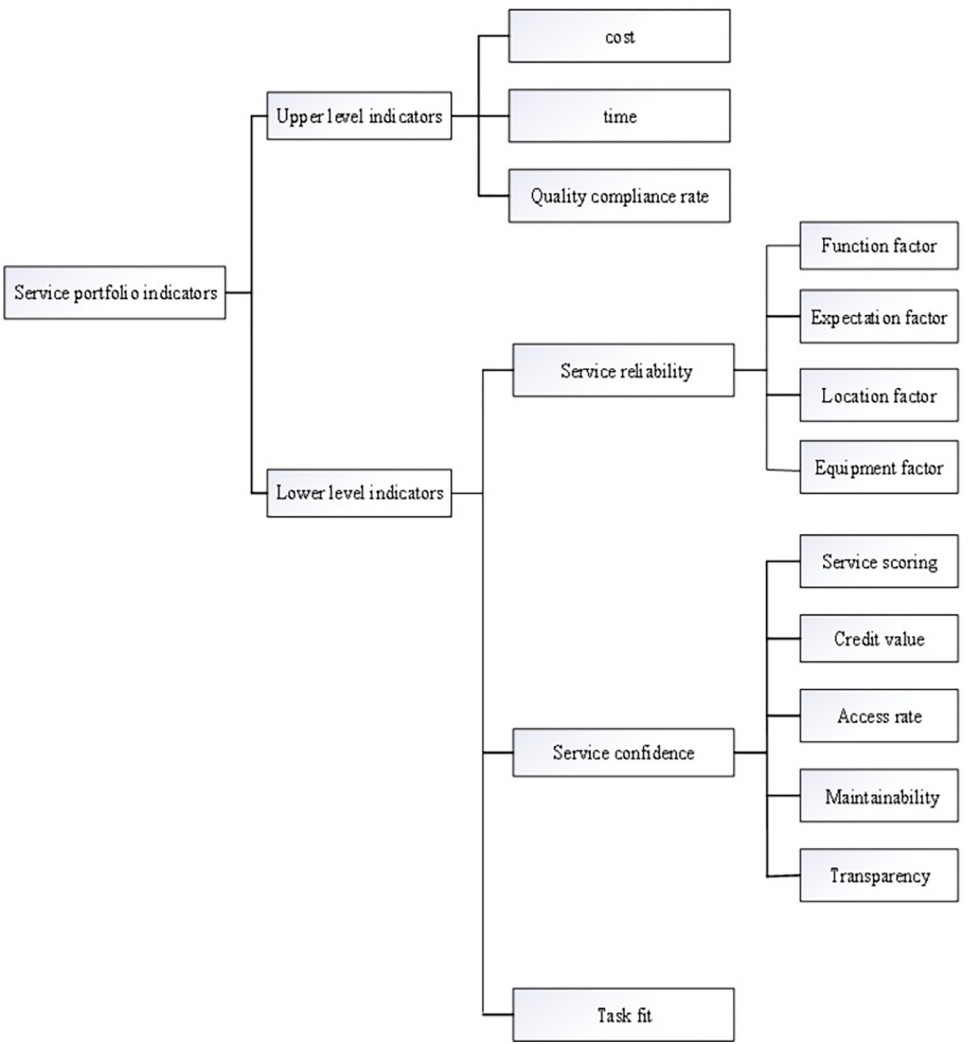

**Fig 2. Relationship between indicators for shared manufacturing services.**

The time cost is mostly comprised of the shared manufacturing service's running time $T_{ir}(R_{ij})$ and the logistics and transportation time $T_{xi}(R_{ij}, R_{(i+1)k})$.

The formula is as follows:

$$T = T_{ir} + T_{xi} = \sum_{i=1}^{m} \sum_{j=1}^{n_i} x_{ij} T_{ir}(R_{ij}) + \sum_{i=1}^{m} \sum_{j=1}^{n_i} \sum_{k=1}^{n_{i+1}} x_{ij,(i+1)k} T_{xi}(R_{ij}, R_{(i+1)k}) \tag{2}$$

**(3) Quality compliance rate**

The quality of the shared manufacturing service shows the quality compliance rate of manufacturing tasks, which represents how well the product's quality passes.

The formula is as follows:

$$Q = \sum_{i=1}^{m} \sum_{j=1}^{n_i} x_{ij} Q_{se}(R_{ij}) / m \tag{3}$$

**Table 1. Evaluation indicators and descriptions of indicators.**

| Primary indicators | Secondary indicators | Indicator Description |
|---|---|---|
| $C$ | $C_{ir}(R_{ij})$ | The cost of raw material consumption, mechanical equipment costs, and labor salaries during the processing process |
| | $C_{xi}(R_{ij}, R_{(i+1)k})$ | The logistics cost of transporting to the next task point after completing a manufacturing task, including labor costs, processing costs, and transportation costs |
| $T$ | $T_{ir}(R_{ij})$ | The actual completion time of shared manufacturing tasks and the time of invalid tasks |
| | $T_{xi}(R_{ij}, R_{(i+1)k})$ | Time for resources to be transferred from the current mission point to the next mission point |
| $Q$ | $Q_{se}(R_{ij})$ | The quality compliance rate represents the qualification status of the product's quality |
| $SR$ | $SR_{FF}(R_{ij})$ | Indicates the frequency, status, and activity of service execution over some time |
| | $SR_{FF}(R_{ij})$ | Indicates the idle status of manufacturing resources |
| | $SR_{LF}(R_{ij})$ | Indicates the distance between the positions of the supply and demand parties |
| | $SR_{QF}(R_{ij})$ | Represent the progressiveness of manufacturing equipment |
| $SC$ | $SR_i$ | The demand side will rate the ith manufacturing task completed |
| | $HV_i$ | The integrity of the demand side to the service provider for the ith manufacturing task is marked on the shared platform |
| | $IR_i$ | Access rate of the ith shared manufacturing service |
| | $MA_i$ | Indicates the maintainability of the ith shared manufacturing task |
| | $PT_i$ | Indicates transparency in the process of completing the ith shared manufacturing task |
| $TC$ | - | Indicates the degree of cooperation between two manufacturing tasks |

## 3.4. Lower-level optimization metrics

**3.4.1. Service reliability modeling.** Service reliability is the probability and ability of a shared manufacturing service to be completed successfully within a given time frame. The influencing factors of service reliability include the execution of the service, the number of executions, the service activity in the past period, the level of manufacturing technology, the utilization rate of idle resources, the distance between the location of manufacturing resources and manufacturing tasks, and the sophistication of the equipment. Based on the similarity between the influencing factors, the influencing factors of service reliability are summarized as function, expectation, location, and equipment factors.

**(1) Function factor: $SR_{FF}(R_{ij})$**

The function factor $SR_{FF}(R_{ij})$ reflects the ith shared service's technical capability to complete the jth manufacturing task, $1 \leq i \leq n$, $1 \leq j \leq m$. $SR_{FF}(R_{ij})$ has a value interval of [0,1]. The greater the execution frequency and service activity is, as well as the total number of tasks completed, the greater the value of technical competence $SR_{FF}(R_{ij})$ is, and vice versa. If the functional match between service and task is inconclusive, $SR_{FF}(R_{ij})$ equals 0.

**(2) Expectation factor: $SR_{EF}(R_{ij})$**

The idle rate of manufacturing resources is defined as the proportion of underused manufacturing resources to total shared manufacturing resources. The higher the idle rate of manufacturing resources is, the higher the expectation of completing manufacturing tasks is, so the degree of acceptance of shared manufacturing services and the expectation of completing manufacturing tasks can be evaluated using the expectation factor. $SR_{EF}(R_{ij})$ denotes the objective state measure of the ith shared service undertaking and completing the jth manufacturing task, $1 \leq i \leq n$, $1 \leq j \leq m$, $SR_{EF}(R_{ij}) \in [0,1]$. The greater idle rate of

manufacturing resources mapped by shared manufacturing service is, the greater the expectation degree of accepting manufacturing tasks is, and therefore the greater the value of expectation factor $SR_{EF}(R_{ij})$ is, and vice versa.

**(3) Location factor:** $SR_{LF}(R_{ij})$

In general, a shorter distance between locations can minimize logistics distribution time while simultaneously lowering execution costs, making it more suitable to the execution of shared manufacturing services. Location Factor is introduced to describe the degree of influence of the relative distance between manufacturing resources mapped by shared manufacturing service and the demand side on the execution of shared manufacturing service. $SR_{LF}(R_{ij})$ denotes the distance factor between the ith shared manufacturing service mapped manufacturing resource and the jth demand side, $1 \leq i \leq n$, $1 \leq j \leq m$, $SR_{LF}(R_{ij}) \in [0,1]$. The smaller the relative distance is, the larger the value of the location factor $SR_{LF}(R_{ij})$ is, and vice versa.

**(4) Equipment factor:** $SR_{QF}(R_{ij})$

Advanced equipment can boost service efficiency, lower labor expenses, and save time on service execution. As a result, in the execution of shared manufacturing services, the equipment factor becomes a constraint. The equipment factor is presented to describe the impact of sophisticated equipment on the execution of shared manufacturing services during the shared manufacturing service process. $SR_{QF}(R_{ij})$ denotes the Equipment Factors metric between the ith shared manufacturing service and the jth demand side, $1 \leq i \leq n$, $1 \leq j \leq m$, $SR_{QF}(R_{ij}) \in [0,1]$. The higher the level of equipment, the greater the value of $SR_{QF}(R_{ij})$, and vice versa.

The reliability indicator can be calculated from the function, expectation, location and equipment factors. The following is the precise formula:

$$SR = \alpha_1 \times SR_{FF}(R_{ij}) + \alpha_2 \times SR_{EF}(R_{ij}) + \alpha_3 \times SR_{LF}(R_{ij}) + \alpha_4 \times SR_{QF}(R_{ij}) \tag{4}$$

Where: $\alpha_1 + \alpha_2 + \alpha_3 + \alpha_4 = 1$, $\alpha_1, \alpha_2, \alpha_3, \alpha_4$, are the weight coefficients of the influencing factors.

**3.4.2. Service confidence modeling.** Service trustworthiness refers to the total evaluation of service transaction behaviors for shared manufacturing services, service composition, service suppliers, service demanders, and shared manufacturing platforms during a specific time period. The service score of shared manufacturing, the recommendation record, the record of lost trust, the number of visits to the service, the number of accidents successfully handled, and the openness and transparency of the service process can all be used to calculate service confidence.

**(1) Service rating:**

After using the service, the demand side of the shared service can rate the manufacturing service on the shared manufacturing platform. The demand side rating interval ranges from 0 to 10 points. The rating value of the shared manufacturing service is calculated by the shared manufacturing platform based on the ratings used by all demand sides. The following is the precise formula:

$$SR_i = \frac{\sum_{g=1}^{E_i}(US_{g,i})}{10E_i} \tag{5}$$

$SR_i$ denotes the ith shared manufacturing service's standardized average demand side rating; $US_{g,i}$ denotes the rating of the gth demand side for the ith shared manufacturing service; and $E_i$ denotes the total number of evaluating demand sides for the ith shared manufacturing service. The greater the service rating is, the more credible the shared service is.

**(2) Credit value:**

After using a shared service, the demand side can submit feedback on the integrity level of that manufacturing service on the shared manufacturing platform. If the manufacturing

service has a breach of trust, the demand side can label it as such when evaluating it; if the demand side believes that the service enjoyed is worth promoting, it can indicate it as such. The credit value is calculated by the shared manufacturing platform based on the breach of trust and recommendation records of the shared manufacturing service. The following is the precise formula:

$$HV_i = \frac{HR_i}{HR_i + LR_i} \tag{6}$$

$HV_i$ denotes the credit value of the ith shared service; $HR_i$ denotes the number of recommended records of the ith shared manufacturing service, and $LR_i$ denotes the number of discredited records of the ith shared manufacturing service. The higher the credit value of the shared service is, the more trustworthy it is.

**(3) Access rate**:

The greater the access rate of a shared manufacturing service is, the greater the number of shared manufacturing demand-side parties accessing the service is, which indicates that the service is trustworthy. Therefore, this paper adopts the access rate as the evaluation index of shared manufacturing service trustworthiness. The following is the precise formula:

$$IR_i = \frac{WH_i}{WH_{\max}} \tag{7}$$

$IR_i$ denotes the ith shared manufacturing service's access rate; $WH_i$ denotes the number of accesses to the ith shared manufacturing service; $WH_{max}$ denotes the number of accesses to the service with the highest access rate among services of the same type as the ith shared manufacturing service.

**(4) Maintainability**:

The higher the number of times the service successfully handles accidents is when the demander of a shared manufacturing service uses it, the more it indicates that the service is trustworthy. Thus, the demander's chance of choosing it increases. Therefore, this paper adopts maintainability to evaluate the trustworthiness of shared manufacturing services. The following is the precise formula:

$$MA_i = \frac{MF_i}{TF_i} \tag{8}$$

$MA_i$ denotes the maintainability of the ith shared manufacturing service; $MF_i$ denotes the number of accidents successfully handled by the ith shared manufacturing service; $TF_i$ denotes the total number of accidents handled by the ith shared manufacturing service.

**(5) Transparency**:

To avoid opportunistic conduct, the entire sharing process must be fully open, ensuring the interests of both the supply and demand sides as well as the fairness of sharing. The shared manufacturing demand side assigns a score from 0 to 5 to the transparency of completed manufacturing tasks on the shared manufacturing platform. The more transparent the shared manufacturing demand side is in using the service, the more trustworthy it is. As a result, the indicator of transparency is used in this study to assess the trustworthiness of the service. It has the following formula:

$$PT_i = \frac{\sum_{d=1}^{d_f} TN_{d,j}}{5d_f} \tag{9}$$

$PT_i$ denotes the transparency of the ith shared manufacturing service; $TN_{d,j}$ denotes the

transparency rating of the jth shared manufacturing task by the dth shared manufacturing demander, and $d_f$ denotes the total number of demanders evaluating the service.

The confidence indicator can be calculated from the service rating, credit value, access rate, maintainability and transparency. The following is the formula:

$$SC = \beta_1 \times SR_I + \beta_2 \times HV_I + \beta_3 \times IR_I + \beta_4 \times MA_I + \beta_5 \times PT_I \tag{10}$$

where $\beta_1 + \beta_2 + \beta_3 + \beta_4 + \beta_5 = 1$, $\beta_1, \beta_2, \beta_3, \beta_4, \beta_5$, are the weight coefficients of the influencing factors.

**3.4.3. Task fit modeling.**   Task fit reflects the degree of cooperation between two shared manufacturing services in a service portfolio. In the service portfolio, the easier the information exchange between shared manufacturing services is, the smoother the material transportation is, and the shorter the time to complete the manufacturing task is, and the higher the task fit between shared manufacturing services is. The task fit in shared manufacturing services is directly reflected in the execution time. Task fit can be assessed by calculating the time required for shared manufacturing services to complete a manufacturing task. For example, the task fit between shared manufacturing services $S_i$ and $S_j$ used to complete manufacturing tasks $J_i$ and $J_j$ can be calculated as follows:

$$TC = \frac{T_i + T_j}{T_{ij}} \tag{11}$$

$T_i$ is the time spent by the shared manufacturing service $S_i$ on its own to complete the manufacturing task $J_i$; $T_j$ is the time spent by the shared manufacturing service $S_j$ on its own to complete the manufacturing task $J_j$; and $T_{ij}$ is the total time spent by the two shared manufacturing services $S_i$ and $S_j$ cooperating to complete the two manufacturing tasks $J_i$ and $J_j$.

## 3.5. A bi-level programming model for service composition optimization of shared manufacturing

The following is a bi-level programming model for service composition optimization in a shared manufacturing environment:

$$(\mathbf{U}) \quad maxF = \left( \frac{C_{max}}{C}, \frac{T_{max}}{T}, \frac{Q}{Q_{min}} \right)^T \tag{12}$$

$$\mathbf{s.t.} \quad C \leq C_{max};$$

$$T \leq T_{max};$$

$$Q_{min} \leq Q;$$

$$(\mathbf{L}) \quad maxf = \omega_1 \times SR + \omega_2 \times SC + \omega_3 \times TC \tag{13}$$

$$\mathbf{s.t.} \; SR \geq SR_{min};$$

$$SC \geq SC_{min};$$

$$TC \geq TC_{min};$$

The upper-level objective function's constraints are passed to the lower-level objective function, which optimizes its objective under the upper-level objective function's constraints and

passes its optimal solution to the upper-level, which further optimizes based on the upper-level objective function to obtain the model's optimal solution. The shared manufacturing service composition bi-level programming model uses the service supplier's demand as the upper-level optimization objective and the service demander's demand as the lower-level optimization objective. This protects the service supplier's interests and gives the service demander the chance to think about different aspects of the manufacturing resources and services, which keeps the manufacturing tasks moving smoothly.

## 4. Model algorithm

### 4.1. Determination of indicator weights—CRITIC method

Diakoulaki [32] proposed the Criteria Importance Though Intercrieria Correlation (CRITIC method) to assign weights to evaluate indicators objectively. The method revolves around two aspects of weighting indicators: contrast and ambivalence.

Its core idea is to calculate objective weights for indicators using two key notions. The first one is the degree of contrast, which is measured by the standard deviation. This shows how big the difference is between the values of the different evaluation programs for the same indicator. In other words, the bigger the standard deviation is, the bigger the difference between the values of the different programs for the same indicator is. The second issue is the disagreement between the evaluation indicators. The conflict between the indicators is based on their correlation, such as the two indicators having a large positive correlation, suggesting a minimal conflict between the two indicators.

The steps are as follows:

**Step 1.** Positive and standardized indicators

There are m objects to be evaluated and n evaluation indicators. It can form a data matrix $X = (X_{ij})m^*n$. Let the elements in the data matrix, after the index normalization processed elements are $X\prime_{ij}$.

If $X_j$ is a negative indicator (smaller and better indicator)

$$X'_{ij} = \frac{\max(x_j) - x_{ij}}{\max(x_j) - \min(x_j)} \tag{14}$$

If $X_j$ is a positive indicator (larger and better indicator)

$$1\ X'_{ij} = \frac{x_{ij} - \min(x_j)}{\max(x_j) - \min(x_j)} \tag{15}$$

**Step2.** Calculate information carrying capacity

Contrast: The standard deviation is used to express the jth indicator's contrastivity.

$$\sigma_j = \sqrt{\frac{\sum_{i=1}^{m}(x'_{ij} - \bar{x'_j})}{m-1}} \tag{16}$$

Contradiction: This reflects the degree of correlation between different indicators. If presenting a significant positive correlation, the smaller the value of contradiction is. Let the size of the contradiction between indicator j and the rest of the indicators be $f_j$.

$$f_j = \sum_{i=1}^{m}(1 - r_{ij}) \tag{17}$$

$r_{ij}$ denotes the correlation coefficient between indicator i and indicator j. The Pearson correlation coefficient, a linear correlation coefficient, is used here.

Information carrying capacity: Let the information carrying capacity of the jth indicator be $C_j$.

**Step3.** Calculate weights

$$\omega_j = \frac{C_j}{\sum_{j=1}^{n} C_j} \tag{18}$$

## 4.2. Improved NSGA-II algorithm

The traditional NSGA-II algorithm has some defects, such as poor convergence speed [33]. Therefore, in this paper, the model is solved by using the improved NSGA-II algorithm, which is improved in three aspects: parent selection, crossover and mutation methods:

**(1) Parent-generation selection:** The original selection method used in the parent-generation selection method is tournament selection. The tournament selection process has certain drawbacks as well. From the standpoint of tournament implementation, the selection of the parent generation is similar to random selection, which impacts the speed of evolution in the population's overall evolutionary process, and the searchability of its evolutionary process cannot be guaranteed. In order to ensure the convergence speed of the algorithm, in the early stage of the algorithm, to ensure the algorithm's searchability, to expand the search space, and to avoid falling into the local optimum; late in the stage, to increase the pressure of the parent generation selection and to ensure the algorithm's convergence so that the algorithm can be better approximated to the actual value, which leads to the selection of the parent generation selection based on the linear ranking method [34].

**(2) Crossover method:** The results of the polynomial crossover method can be understood as randomly varying values within a specific range, which does not necessarily guarantee that the child individuals are better than the parent individuals. This situation also leads to a slow algorithm convergence speed. To address this situation and boost the algorithm's convergence speed, a novel crossover strategy based on the particle swarm algorithm is developed. The proposed new crossover method, guided crossover, ensures that the evolutionary direction of offspring generated by each hybridization is determined based on the optimal direction between the two parent individuals and the two parent individuals' common genes, thereby increasing the algorithm's convergence speed so that the generated individuals evolve in the best possible direction.

**(3) Mutation approach:** To associate the role of the mutation operator with the number of generations, the algorithm can be mutated more prominently early in the mutation, and as the number of generations of evolution increases, the range of the mutation shrinks and shrinks, increasing the algorithm's ability to be fine-tuned. Z. Michalewicz's non-uniform variation is employed [35]. Non-uniform mutations can be explored throughout the entire parameter space in the initial iteration of genetic algorithms, enhancing population diversity and enabling the algorithm to discover a larger solution space. With an increasing number of iterations, non-uniform mutations do localized searches around individuals, hence enhancing the evolutionary rate of the algorithm. Non-uniform mutation operators have an algebraic function, leading to a broader spectrum of mutations during the initial stages of the process. The scope of mutations decreases as the number of evolutionary generations increases. This enhances the algorithm's ability to fine-tune and enables it to accurately identify the optimal solution.

The steps are as follows:

**Step1.** In order to construct chromosome genes corresponding to individuals in the population, candidate manufacturing resource services are encoded by using integer coding. As shown in Fig 3, a shared manufacturing service composition can be represented as a chromosomal section having n genes. For example, for the candidate service composition {$MMR_{1,1}$,

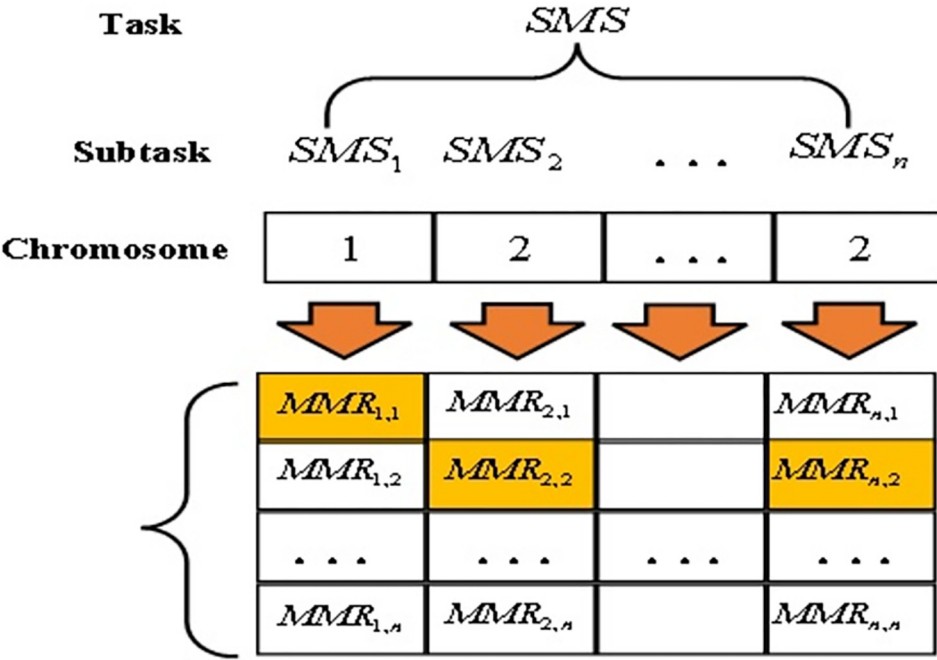

**Fig 3. Service composition and chromosome mapping relationship diagram.**

$MMR_{2,3},MMR_{3,1},MMR_{4,2},MMR_{5,3}\}$, the corresponding chromosome encoding method is [1, 3, 1, 2, 3].

**Step2.** Create an initialized population at random with a total number of individuals as NP and compute the objective function value, then do a rapid, non-dominated ranking of the initialized population and compute the individual crowding degree after stratification.

**Step3.** Individuals in the population are ordered in descending order of rank and ascending order of crowding distance in linear selection ranking, from lowest to greatest adaptability value. The higher the selection probability is, the lower the rank (the larger the adaptation value) is. Then, to obtain the parent population, assign the selection probability of each individual according to some relevant linear functions.

**Step4.** Bootstrap crossover and non-uniform mutation operations are carried out based on the set crossover and mutation probabilities to generate an offspring population with the same number of individuals as the initial population.

**Step5.** Combine the parent and offspring populations to obtain a combined population of 2NP individuals.

**Step6.** Perform quick non-dominated sorting on the combined population, determine the crowding degree of individuals, calculate the individuals' fitness based on the non-dominated rank and crowding degree, and keep the optimal NP individuals by the elite retention strategy to generate a new offspring population.

**Step7.** Make *inter = inter* + 1, repeat Step3~ Step6 until the maximum number of genetic generations 180 is reached, and then continue Step8.

**Step8.** Find the pareto solution set of the upper-level programming objective function.

**Step9.** Bring the upper-level objective function's pareto solution set into the lower- level model and use the optimal value as the bi-level programming model's final solution.

The whole algorithm flow is shown in Fig 4.

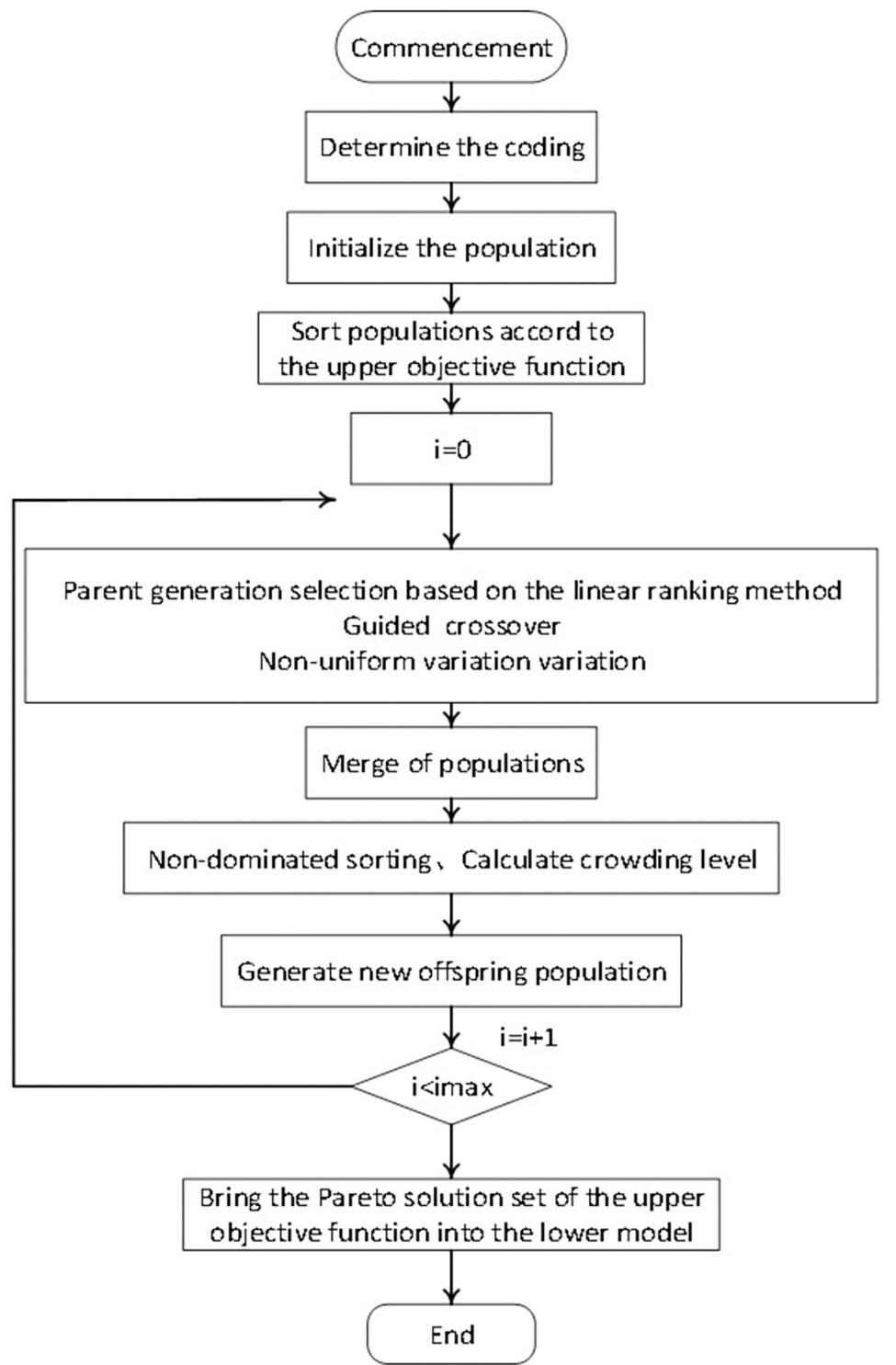

**Fig 4. Improved NSGA-II algorithm flowchart.**

## 5. Numerical example analysis

### 5.1. Algorithm testing

This paper uses generation distance (GD) and inverse generation distance (IGD) as evaluation metrics to compare the performance of the improved NSGA-II algorithm with that of the traditional NSGA-II algorithm. The generation distance is the distance between the algorithm-generated solution set and the ideal solution. It reflects the difference between the average performance of people in the population and the ideal answer as the algorithm evolves. The lower the GD value is, the more efficient and faster the algorithm is at finding people near the ideal solution in a short period of evolutionary generations, which indicates the algorithm's efficiency and convergence speed. The inverse generational distance (IGD) assesses an algorithm's convergence and distribution performance by determining the lowest distance between each point on the genuine Pareto front and the set of individuals produced by the method. Better convergence and distribution properties result from the algorithm's ability to locate and approach points on the true Pareto front more accurately when the IGD value is lower. This implies a smaller deviation between the set of solutions generated by the algorithm and the optimal set of solutions.

Set the population size NP = 200 and adjust the crossover and mutation probabilities to pc = 0.7 and pm = 0.03. Select three test functions, DTLZ1, DTLZ2, and DTLZ3, and test the GD and IGD values of the traditional NSGA-II algorithm and improved NSGA-II algorithm for 100, 300, and 500 iterations, respectively. Test each function 10 times and take the average value, and record the algorithm running time for different iterations. The experimental results are shown in Tables 2 and 3.

As the number of iterations increases, the convergence and diversity of the algorithm will be affected. According to Tables 2 and 3, it can be seen that the GD and IGD values of all test functions of the improved NSGA-II algorithm are smaller than those of the traditional NSGA-II algorithm. This indicates that the improved NSGA-II algorithm can approach the real optimal solution set faster during the iteration process, effectively utilize the search space, and obtain diverse solutions with faster convergence speed. The running time of the improved NSGA-II algorithm for solving is 69.99% less than that of traditional NSGA-II algorithm.

### 5.2. Sensitivity analysis

In order to analyze the impacts of crossover and mutation probabilities on the solution results, while keeping other parameters constant, crossover probability = [0.1, 0.3, 0.5, 0.7] and mutation probability = [0.01, 0.02, 0.03, 0.04] were taken. The improved NSGA-II algorithm was run 10 times, and the required number of iterations to converge to the optimal solution was

**Table 2. GD and IGD values of the improved NSGA-II algorithm.**

| Iterations | Test the function | Improved GD | Improved IGD | Run time |
|---|---|---|---|---|
| 100 | DTLZ1 | 0.038148 | 0.059894 | 13.175 |
| | DTLZ2 | 1.091276 | 0.096658 | |
| | DTLZ3 | 0.594196 | 0.072488 | |
| 300 | DTLZ1 | 0.486567 | 0.042839 | 38.242 |
| | DTLZ2 | 0.42805 | 0.06704 | |
| | DTLZ3 | 0.505587 | 0.03498 | |
| 500 | DTLZ1 | 0.507215 | 0.052317 | 63.836 |
| | DTLZ2 | 0.509513 | 0.096307 | |
| | DTLZ3 | 0.487372 | 0.072732 | |

**Table 3. GD and IGD values of the traditional NSGA-II algorithm.**

| Iterations | Test the function | GD | IGD | Run time |
|---|---|---|---|---|
| 100 | DTLZ1 | 2.58422 | 0.133574 | 44.262 |
| | DTLZ2 | 2.44801 | 0.114911 | |
| | DTLZ3 | 2.58241 | 0.099055 | |
| 300 | DTLZ1 | 2.43341 | 0.12879 | 135.591 |
| | DTLZ2 | 2.55335 | 0.12061 | |
| | DTLZ3 | 2.60516 | 0.116259 | |
| 500 | DTLZ1 | 2.46026 | 0.119588 | 212.699 |
| | DTLZ2 | 2.51608 | 0.133373 | |
| | DTLZ3 | 2.47621 | 0.128891 | |

recorded and averaged. The result was shown in the Figs 5 and 6. From the graph, it can be seen that when the crossover probability is 0.7 and the mutation probability is 0.3, it can ensure that the improved NSGA-II converges to the optimal solution at the fastest speed.

## 5.3. Data preparation

Using a manufacturing task released by a specific manufacturing company as an example, in the calculation example, the shared manufacturing demand submits the manufacturing task and related QoS attribute needs on the shared manufacturing platform. As illustrated in Table 4, the platform divides the manufacturing task into four sub-tasks, each with its own corresponding candidate manufacturing service. Tables 5 and 6 display the relevant values of the

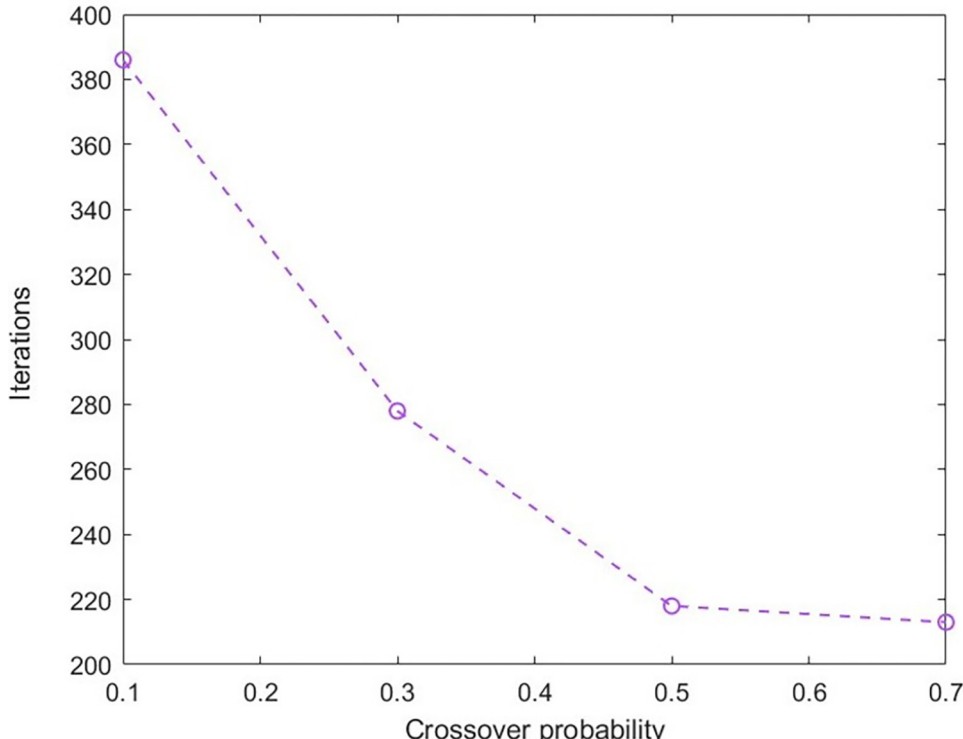

**Fig 5. The number of iterations required to converge to the optimal solution under different crossover probabilities.**

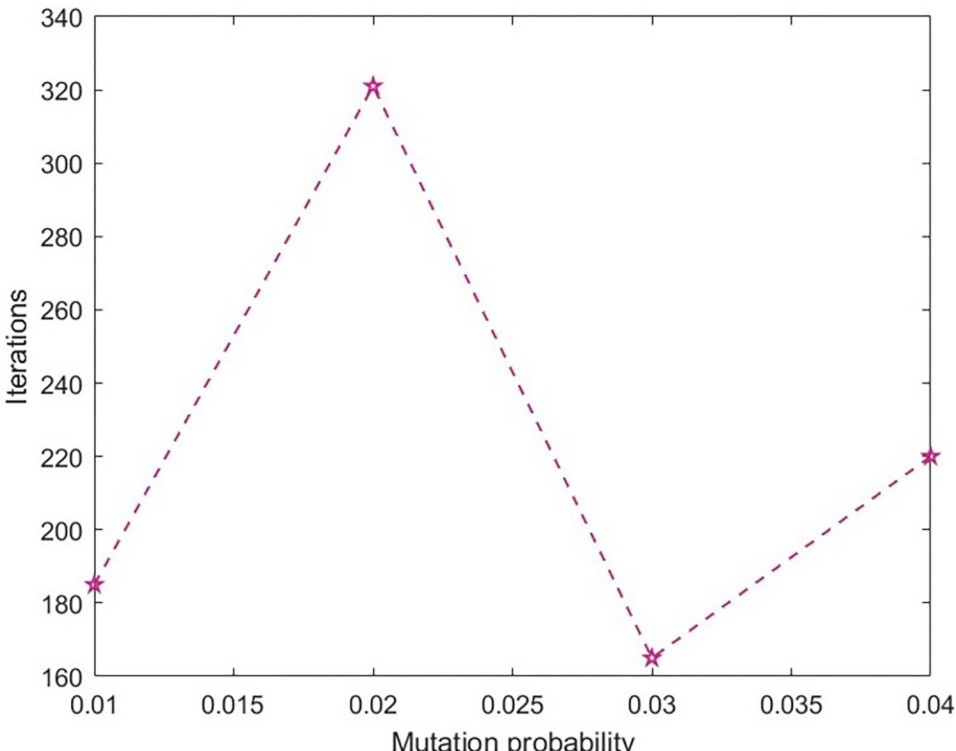

**Fig 6. The number of iterations required to converge to the optimal solution under different crossover probabilities.**

candidate manufacturing service evaluation indicators for the upper and lower objective functions.

The CRITIC method is used to determine the indicator weights of the lower-level objective function. Since all values are as large as possible in the lower-level objective function of this article, there is no need to deal with negative indicators. All indicators need to be standardized and multiplied by the contrast intensity and conflict, and then the information-carrying capacity is calculated. Finally, the weights are calculated, and the larger the information-carrying capacity is, the greater the weight is. The specific parameters of the model are shown in Table 7.

## 5.4. Example solution

This paper uses the improved NSGA-II algorithm to solve the model in the MATLAB 2022b computing environment (running on 12th Gen Intel (R) Core (TM) i5-12500H, 2.50 GHz, Windows 11 Home Chinese version, RAM16.0GB), with an initial population NP = 200,

**Table 4. Candidate manufacturing resource service set.**

| Manufacturing sub-tasks (*SMS*) | $SMS_1$ | $SMS_2$ | $SMS_3$ | $SMS_4$ | $SMS_5$ |
|---|---|---|---|---|---|
| Candidate Manufacturing Resource Services (*MMR*) | $MMR_{1,1}$ | $MMR_{2,1}$ | $MMR_{3,1}$ | $MMR_{4,1}$ | $MMR_{5,1}$ |
| | $MMR_{1,2}$ | $MMR_{2,2}$ | $MMR_{3,2}$ | $MMR_{4,2}$ | $MMR_{5,2}$ |
| | $MMR_{1,3}$ | $MMR_{2,3}$ | $MMR_{3,3}$ | $MMR_{4,3}$ | $MMR_{5,3}$ |
| | $MMR_{1,4}$ | – | – | $MMR_{4,4}$ | – |

**Table 5. Candidate manufacturing service upper-level objective function parameters.**

| | $C_{ir}$ | $C_{xi}$ | | $T_{ir}$ | $T_{xi}$ | | $Q_{se}$ |
|---|---|---|---|---|---|---|---|
| $MMR_{1,1}$ | 347 | $MMR_{2,1}$ | 56 | 23 | $MMR_{2,1}$ | 29 | 0.94 |
| | | $MMR_{2,2}$ | 22 | | $MMR_{2,2}$ | 28 | |
| | | $MMR_{2,3}$ | 36 | | $MMR_{2,3}$ | 10 | |
| $MMR_{1,2}$ | 296 | $MMR_{2,1}$ | 84 | 43 | $MMR_{2,1}$ | 35 | 0.98 |
| | | $MMR_{2,2}$ | 22 | | $MMR_{2,2}$ | 23 | |
| | | $MMR_{2,3}$ | 27 | | $MMR_{2,3}$ | 32 | |
| $MMR_{1,3}$ | 389 | $MMR_{2,1}$ | 14 | 35 | $MMR_{2,1}$ | 20 | 0.96 |
| | | $MMR_{2,2}$ | 33 | | $MMR_{2,2}$ | 16 | |
| | | $MMR_{2,3}$ | 34 | | $MMR_{2,3}$ | 36 | |
| $MMR_{1,4}$ | 441 | $MMR_{2,1}$ | 63 | 31 | $MMR_{2,1}$ | 45 | 0.94 |
| | | $MMR_{2,2}$ | 67 | | $MMR_{2,2}$ | 32 | |
| | | $MMR_{2,3}$ | 35 | | $MMR_{2,3}$ | 18 | |
| $MMR_{2,1}$ | 209 | $MMR_{3,1}$ | 17 | 50 | $MMR_{3,1}$ | 47 | 0.92 |
| | | $MMR_{3,2}$ | 24 | | $MMR_{3,2}$ | 39 | |
| | | $MMR_{3,3}$ | 24 | | $MMR_{3,3}$ | 12 | |
| $MMR_{2,2}$ | 200 | $MMR_{3,1}$ | 27 | 24 | $MMR_{3,1}$ | 35 | 0.93 |
| | | $MMR_{3,2}$ | 47 | | $MMR_{3,2}$ | 46 | |
| | | $MMR_{3,3}$ | 47 | | $MMR_{3,3}$ | 26 | |
| $MMR_{2,3}$ | 210 | $MMR_{3,1}$ | 29 | 46 | $MMR_{3,1}$ | 37 | 0.97 |
| | | $MMR_{3,2}$ | 46 | | $MMR_{3,2}$ | 34 | |
| | | $MMR_{3,3}$ | 37 | | $MMR_{3,3}$ | 20 | |
| $MMR_{3,1}$ | 309 | $MMR_{4,1}$ | 23 | 12 | $MMR_{4,1}$ | 17 | 0.98 |
| | | $MMR_{4,2}$ | 25 | | $MMR_{4,2}$ | 20 | |
| | | $MMR_{4,3}$ | 25 | | $MMR_{4,3}$ | 40 | |
| | | $MMR_{4,4}$ | 34 | | $MMR_{4,4}$ | 37 | |
| $MMR_{3,2}$ | 195 | $MMR_{4,1}$ | 11 | 20 | $MMR_{4,1}$ | 31 | 0.98 |
| | | $MMR_{4,2}$ | 30 | | $MMR_{4,2}$ | 34 | |
| | | $MMR_{4,3}$ | 40 | | $MMR_{4,3}$ | 28 | |
| | | $MMR_{4,4}$ | 51 | | $MMR_{4,4}$ | 17 | |
| $MMR_{3,3}$ | 390 | $MMR_{4,1}$ | 54 | 30 | $MMR_{4,1}$ | 25 | 0.99 |
| | | $MMR_{4,2}$ | 13 | | $MMR_{4,2}$ | 25 | |
| | | $MMR_{4,3}$ | 39 | | $MMR_{4,3}$ | 10 | |
| | | $MMR_{4,4}$ | 41 | | $MMR_{4,4}$ | 28 | |
| $MMR_{4,1}$ | 195 | $MMR_{5,1}$ | 33 | 25 | $MMR_{5,1}$ | 40 | 0.91 |
| | | $MMR_{5,2}$ | 38 | | $MMR_{5,2}$ | 25 | |
| | | $MMR_{5,3}$ | 34 | | $MMR_{5,3}$ | 30 | |
| $MMR_{4,2}$ | 221 | $MMR_{5,1}$ | 46 | 10 | $MMR_{5,1}$ | 24 | 0.96 |
| | | $MMR_{5,2}$ | 26 | | $MMR_{5,2}$ | 37 | |
| | | $MMR_{5,3}$ | 34 | | $MMR_{5,3}$ | 30 | |
| $MMR_{4,3}$ | 201 | $MMR_{5,1}$ | 21 | 17 | $MMR_{5,1}$ | 31 | 0.99 |
| | | $MMR_{5,2}$ | 37 | | $MMR_{5,2}$ | 19 | |
| | | $MMR_{5,3}$ | 59 | | $MMR_{5,3}$ | 13 | |
| $MMR_{4,4}$ | 121 | $MMR_{5,1}$ | 39 | 20 | $MMR_{5,1}$ | 13 | 0.95 |
| | | $MMR_{5,2}$ | 38 | | $MMR_{5,2}$ | 25 | |
| | | $MMR_{5,3}$ | 47 | | $MMR_{5,3}$ | 20 | |

(*Continued*)

**Table 5.** (Continued)

| | $C_{ir}$ | $C_{xi}$ | | $T_{ir}$ | $T_{xi}$ | | $Q_{se}$ |
|---|---|---|---|---|---|---|---|
| $MMR_{5,1}$ | 279 | demand | 48 | 32 | demand | 33 | 0.98 |
| | | Demand | 15 | | demand | 20 | |
| | | Demand | 57 | | demand | 26 | |
| $MMR_{5,2}$ | 254 | demand | 17 | 20 | demand | 26 | 0.90 |
| | | Demand | 34 | | demand | 34 | |
| | | Demand | 64 | | demand | 30 | |
| $MMR_{5,3}$ | 259 | Demand | 45 | 21 | demand | 35 | 0.95 |
| | | Demand | 42 | | demand | 38 | |
| | | Demand | 14 | | demand | 19 | |

maximum iteration interval = 180, guided crossover probability pc = 0.7, and non-uniform mutation probability pm = 0.03. These parameters were referenced from earlier literature utilizing genetic algorithms and determined by continual debugging during the experiment.

As illustrated in Fig 7, we calculate the pareto frontier of the optimal solution set consisting of 200 solutions. The figure shows that each point is a non-dominated service composition scheme. The whole pareto optimal solution set is on the first level pareto frontier surface and spread out evenly, making it an ideal pareto optimal solution set.

Substitute the upper-level model's pareto solution set into the lower-level model to obtain the bi-level programming model's overall optimal solution set and 200 service composition schemes. Table 8 displays all of the combination schemes derived after deleting duplicate data. Table 9 displays the first five sets of globally optimal service compositions. The shared manufacturing service platform ranks the best solutions based on the superiority and inferiority of candidate service compositions. The results are fed back to the demand side of the shared manufacturing service. Based on their needs, the demand side might select solutions that match the conditions. For example, the top-ranked shared manufacturing service combo solution can be used to perform manufacturing duties.

**Table 6. Candidate manufacturing service lower-level objective function parameters.**

| | $SR$ | $SC$ | $TC$ |
|---|---|---|---|
| $MMR_{1,1}$ | 0.55904 | 0.672078184 | 0.84 |
| $MMR_{1,2}$ | 0.544812 | 0.481346565 | 0.58 |
| $MMR_{1,3}$ | 0.77041 | 0.692402911 | 0.65 |
| $MMR_{1,4}$ | 0.755802 | 0.737583594 | 0.85 |
| $MMR_{2,1}$ | 0.728056 | 0.638531398 | 0.5 |
| $MMR_{2,2}$ | 0.598628 | 0.541347859 | 0.74 |
| $MMR_{2,3}$ | 0.560612 | 0.662251745 | 0.59 |
| $MMR_{3,1}$ | 0.533036 | 0.727140756 | 0.76 |
| $MMR_{3,2}$ | 0.842956 | 0.611902188 | 0.86 |
| $MMR_{3,3}$ | 0.711076 | 0.75604184 | 0.86 |
| $MMR_{4,1}$ | 0.59799 | 0.800091729 | 0.68 |
| $MMR_{4,1}$ | 0.654556 | 0.719671548 | 0.59 |
| $MMR_{4,3}$ | 0.52806 | 0.808139226 | 0.84 |
| $MMR_{4,4}$ | 0.807288 | 0.752215345 | 0.79 |
| $MMR_{5,1}$ | 0.485066 | 0.711120169 | 0.87 |
| $MMR_{5,2}$ | 0.521062 | 0.71296394 | 0.89 |
| $MMR_{5,3}$ | 0.495552 | 0.657693819 | 0.72 |

**Table 7. Model parameter values.**

| Model parameter | Value | Model parameter | Value |
|:---:|:---:|:---:|:---:|
| $\alpha_1$ | 0.3364 | $\omega_1$ | 0.4029 |
| $\alpha_2$ | 0.2516 | $\omega_2$ | 0.2657 |
| $\alpha_3$ | 0.2126 | $\omega_3$ | 0.3314 |
| $\alpha_4$ | 0.1994 | $C_{max}$ | 20000 |
| $\beta_1$ | 0.2071 | $T_{max}$ | 300 |
| $\beta_2$ | 0.1505 | $Q_{min}$ | 0.60 |
| $\beta_3$ | 0.1831 | $SR_{min}$ | 0.40 |
| $\beta_4$ | 0.1992 | $SC_{min}$ | 0.45 |
| $\beta_5$ | 0.2601 | $TC_{min}$ | 0.48 |

## 5.5. Result analysis

Previously, manufacturing service composition optimization strategies mostly focused on service demand indicators while ignoring service provider interests. At the same time, for shared manufacturing service evaluation indicators, there is no more specific definition of service reliability or service confidence. So, this paper suggests a bi-level programming model for the composition of shared manufacturing services. The upper-level model focuses on cost, time, and the rate of quality compliance. The lower-level model focuses on service reliability, service confidence, and task fit. These are evaluation indicators for the demand side of shared manufacturing. Equipment factors are added to the factors affecting service reliability; maintainability and process transparency are added to the factors affecting service confidence, and the improved NSGA-II is used to solve them. To verify the algorithm's effectiveness, the results are compared with those of traditional NSGA-II for solving the upper-level programming model in the same experimental environment.

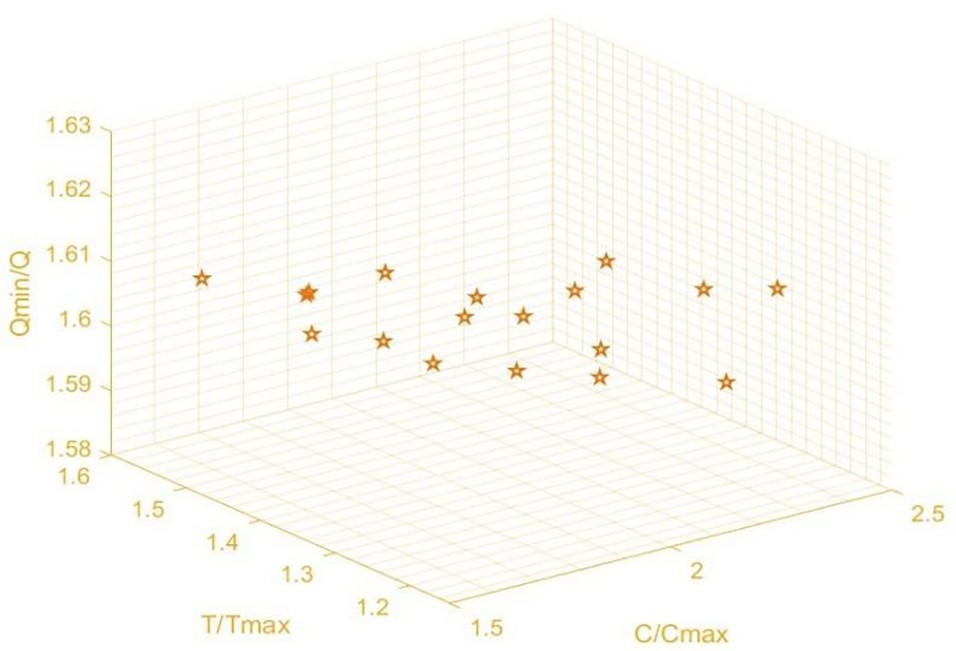

**Fig 7. Upper objective function pareto solution set.**

**Table 8. All candidate service combination schemes.**

| $SMS_1$ | $SMS_2$ | $SMS_3$ | $SMS_4$ | $SMS_5$ | $SMS_1$ | $SMS_2$ | $SMS_3$ | $SMS_4$ | $SMS_5$ |
|---|---|---|---|---|---|---|---|---|---|
| $MMR_{1,4}$ | $MMR_{2,2}$ | $MMR_{3,3}$ | $MMR_{4,2}$ | $MMR_{5,2}$ | $MMR_{1,1}$ | $MMR_{2,2}$ | $MMR_{3,3}$ | $MMR_{4,2}$ | $MMR_{5,3}$ |
| $MMR_{1,4}$ | $MMR_{2,2}$ | $MMR_{3,2}$ | $MMR_{4,2}$ | $MMR_{5,1}$ | $MMR_{1,4}$ | $MMR_{2,3}$ | $MMR_{3,2}$ | $MMR_{4,1}$ | $MMR_{5,3}$ |
| $MMR_{1,4}$ | $MMR_{2,3}$ | $MMR_{3,2}$ | $MMR_{4,2}$ | $MMR_{5,2}$ | $MMR_{1,4}$ | $MMR_{2,3}$ | $MMR_{3,1}$ | $MMR_{4,3}$ | $MMR_{5,2}$ |
| $MMR_{1,4}$ | $MMR_{2,3}$ | $MMR_{3,3}$ | $MMR_{4,2}$ | $MMR_{5,2}$ | $MMR_{1,3}$ | $MMR_{2,1}$ | $MMR_{3,3}$ | $MMR_{4,3}$ | $MMR_{5,3}$ |
| $MMR_{1,4}$ | $MMR_{2,3}$ | $MMR_{3,2}$ | $MMR_{4,2}$ | $MMR_{5,1}$ | $MMR_{1,4}$ | $MMR_{2,3}$ | $MMR_{3,1}$ | $MMR_{4,3}$ | $MMR_{5,1}$ |
| $MMR_{1,4}$ | $MMR_{2,2}$ | $MMR_{3,3}$ | $MMR_{4,3}$ | $MMR_{5,1}$ | $MMR_{1,1}$ | $MMR_{2,3}$ | $MMR_{3,3}$ | $MMR_{4,2}$ | $MMR_{5,3}$ |
| $MMR_{1,4}$ | $MMR_{2,1}$ | $MMR_{3,2}$ | $MMR_{4,2}$ | $MMR_{5,3}$ | $MMR_{1,1}$ | $MMR_{2,2}$ | $MMR_{3,3}$ | $MMR_{4,3}$ | $MMR_{5,3}$ |
| $MMR_{1,3}$ | $MMR_{2,1}$ | $MMR_{3,3}$ | $MMR_{4,2}$ | $MMR_{5,2}$ | $MMR_{1,4}$ | $MMR_{2,3}$ | $MMR_{3,1}$ | $MMR_{4,2}$ | $MMR_{5,3}$ |
| $MMR_{1,4}$ | $MMR_{2,1}$ | $MMR_{3,3}$ | $MMR_{4,2}$ | $MMR_{5,3}$ | $MMR_{1,1}$ | $MMR_{2,1}$ | $MMR_{3,3}$ | $MMR_{4,3}$ | $MMR_{5,3}$ |
| $MMR_{1,4}$ | $MMR_{2,2}$ | $MMR_{3,3}$ | $MMR_{4,1}$ | $MMR_{5,2}$ | $MMR_{1,3}$ | $MMR_{2,3}$ | $MMR_{3,3}$ | $MMR_{4,3}$ | $MMR_{5,3}$ |
| $MMR_{1,4}$ | $MMR_{2,3}$ | $MMR_{3,2}$ | $MMR_{4,2}$ | $MMR_{5,3}$ | $MMR_{1,1}$ | $MMR_{2,3}$ | $MMR_{3,2}$ | $MMR_{4,3}$ | $MMR_{5,3}$ |
| $MMR_{1,4}$ | $MMR_{2,3}$ | $MMR_{3,2}$ | $MMR_{4,2}$ | $MMR_{5,3}$ | $MMR_{1,2}$ | $MMR_{2,2}$ | $MMR_{3,2}$ | $MMR_{4,2}$ | $MMR_{5,1}$ |
| $MMR_{1,1}$ | $MMR_{2,2}$ | $MMR_{3,3}$ | $MMR_{4,2}$ | $MMR_{5,2}$ | $MMR_{1,1}$ | $MMR_{2,2}$ | $MMR_{3,1}$ | $MMR_{4,3}$ | $MMR_{5,2}$ |
| $MMR_{1,4}$ | $MMR_{2,2}$ | $MMR_{3,3}$ | $MMR_{4,3}$ | $MMR_{5,3}$ | $MMR_{1,1}$ | $MMR_{2,3}$ | $MMR_{3,3}$ | $MMR_{4,3}$ | $MMR_{5,3}$ |
| $MMR_{1,3}$ | $MMR_{2,3}$ | $MMR_{3,3}$ | $MMR_{4,2}$ | $MMR_{5,1}$ | $MMR_{1,2}$ | $MMR_{2,2}$ | $MMR_{3,3}$ | $MMR_{4,2}$ | $MMR_{5,1}$ |
| $MMR_{1,3}$ | $MMR_{2,1}$ | $MMR_{3,3}$ | $MMR_{4,3}$ | $MMR_{5,1}$ | $MMR_{1,4}$ | $MMR_{2,3}$ | $MMR_{3,1}$ | $MMR_{4,3}$ | $MMR_{5,3}$ |
| $MMR_{1,1}$ | $MMR_{2,2}$ | $MMR_{3,2}$ | $MMR_{4,3}$ | $MMR_{5,1}$ | $MMR_{1,2}$ | $MMR_{2,3}$ | $MMR_{3,3}$ | $MMR_{4,2}$ | $MMR_{5,2}$ |
| $MMR_{1,3}$ | $MMR_{2,2}$ | $MMR_{3,3}$ | $MMR_{4,2}$ | $MMR_{5,3}$ | $MMR_{1,3}$ | $MMR_{2,3}$ | $MMR_{3,1}$ | $MMR_{4,3}$ | $MMR_{5,1}$ |
| $MMR_{1,3}$ | $MMR_{2,3}$ | $MMR_{3,2}$ | $MMR_{4,3}$ | $MMR_{5,1}$ | $MMR_{1,2}$ | $MMR_{2,2}$ | $MMR_{3,3}$ | $MMR_{4,3}$ | $MMR_{5,2}$ |
| $MMR_{1,4}$ | $MMR_{2,3}$ | $MMR_{3,3}$ | $MMR_{4,3}$ | $MMR_{5,3}$ | $MMR_{1,2}$ | $MMR_{2,2}$ | $MMR_{3,3}$ | $MMR_{4,3}$ | $MMR_{5,2}$ |
| $MMR_{1,4}$ | $MMR_{2,2}$ | $MMR_{3,1}$ | $MMR_{4,3}$ | $MMR_{5,1}$ | | | | | |

Fig 8–10 show that the improved NSGA-II algorithm's cost curve tends to converge after 144 generations and converges faster than the traditional NSGA-II. In the same way, the time and quality compliance rate curves tend to converge in the 140 and 145 generations. Both of these generations converge faster than the traditional NSGA-II algorithm, which proves that the improved NSGA-II algorithm works.

Put the Pareto solution set from the upper-level function into the lower-level function, then compare the best solutions from the standard and improved NSGA-II algorithms. Save the solution sets from iterations 50, 100, 150, and 200, and make curves for each of them.

As seen in Figs 11 and 12, the pictures show that the curve of the traditional NSGA-II algorithm stays unstable after 200 iterations, but the curve of the improved NSGA-II method tends to be stable after 200 iterations. This means that the algorithm is reliable and has found a pretty good solution after several iterations, which proved the algorithm's usefulness.

## 6. Conclusions and future direction

This paper considers a bi-level programming model for the supply and demand sides in the shared manufacturing environment based on the improved NSGA-II. Taking into account the

**Table 9. Candidate service composition.**

| | Candidate Service Portfolio | | | | | Cmax/C | Tmax/T | Q/Qmin | f |
|---|---|---|---|---|---|---|---|---|---|
| 1 | $MMR_{1,4}$ | $MMR_{2,2}$ | $MMR_{3,3}$ | $MMR_{4,2}$ | $MMR_{5,2}$ | 2.1763 | 1.2766 | 1.5733 | 0.7225 |
| 2 | $MMR_{1,4}$ | $MMR_{2,2}$ | $MMR_{3,2}$ | $MMR_{4,2}$ | $MMR_{5,1}$ | 1.8416 | 1.1858 | 1.5967 | 0.7211 |
| 3 | $MMR_{1,4}$ | $MMR_{2,3}$ | $MMR_{3,2}$ | $MMR_{4,2}$ | $MMR_{5,2}$ | 1.9646 | 1.2000 | 1.5833 | 0.7189 |
| 4 | $MMR_{1,4}$ | $MMR_{2,2}$ | $MMR_{3,2}$ | $MMR_{4,2}$ | $MMR_{5,2}$ | 2.2548 | 1.2658 | 1.5867 | 0.7159 |
| 5 | $MMR_{1,4}$ | $MMR_{2,3}$ | $MMR_{3,3}$ | $MMR_{4,2}$ | $MMR_{5,1}$ | 1.8815 | 1.2048 | 1.6100 | 0.7146 |

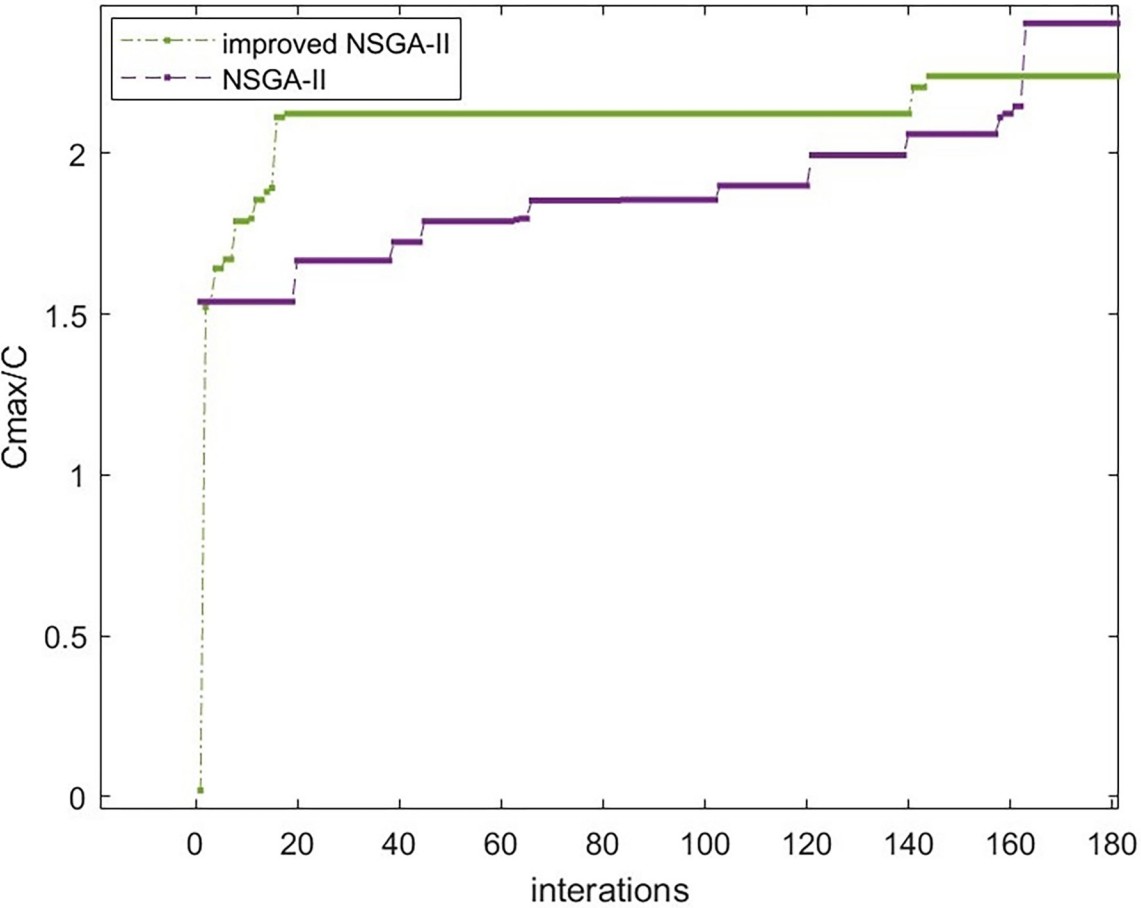

**Fig 8. Comparison of cost convergence curves.**

impact of the progressiveness of equipment provided by the supplier of shared manufacturing services on the completion of manufacturing tasks, the equipment factor is included in the service reliability index in the evaluation index system of shared manufacturing services. Simultaneously, maintainability and transparency have been introduced into the service confidence index, taking into account whether the supplier can successfully handle accidents during the manufacturing process, as well as the openness and transparency provided by the supplier to the demand side in various production and manufacturing processes. The improved NSGA-II method was utilized to solve the model, and its feasibility and effectiveness are demonstrated by comparing the results to the traditional NSGA-II.

This paper mainly considers the research on the service combination optimization method from the perspective of the supply side and the demand side, but in the actual production and manufacturing process, the supply side of the shared manufacturing platform also has its own demand for the indicators of the whole production process. The next step will be to consider the interests of the three parties from the perspective of the shared manufacturing platform and improve the evaluation index system of the shared manufacturing service composition. For example, it is to incorporate the substitutability of services into the evaluation indicators required by the suppliers of shared manufacturing platforms. It is also to consider introducing a time decay function to handle service evaluation indicators.

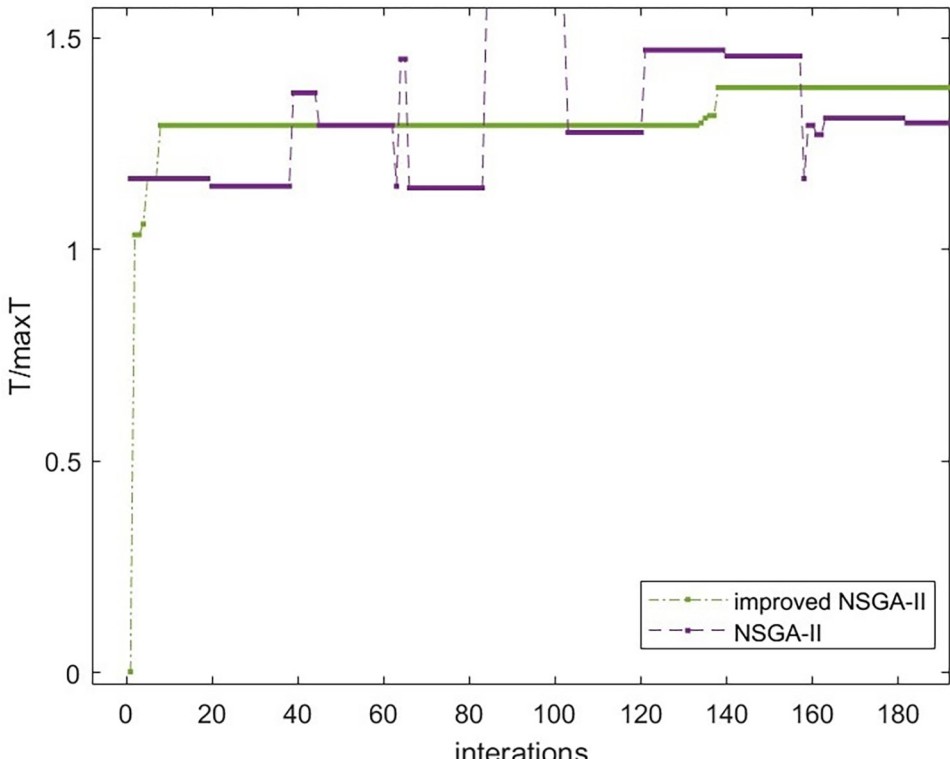

**Fig 9. Comparison of time convergence curves.**

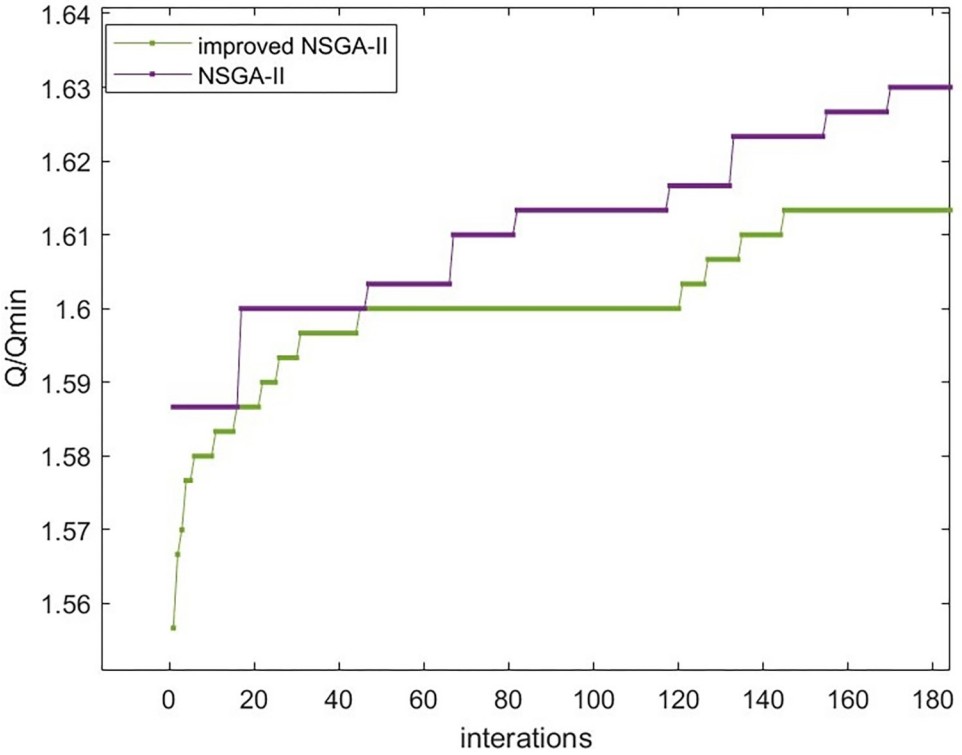

**Fig 10. Comparison of convergence curves for quality compliance rate.**

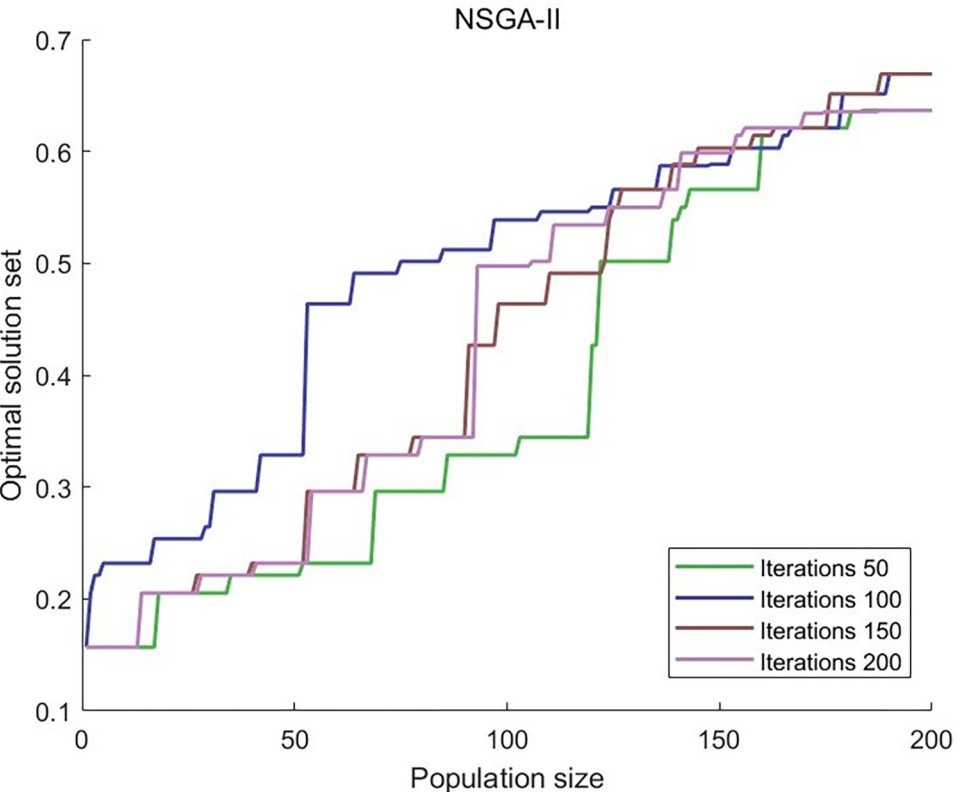

**Fig 11. Using NSGA-II to obtain the optimal solution for iterations of 50, 100, 150, and 200.**

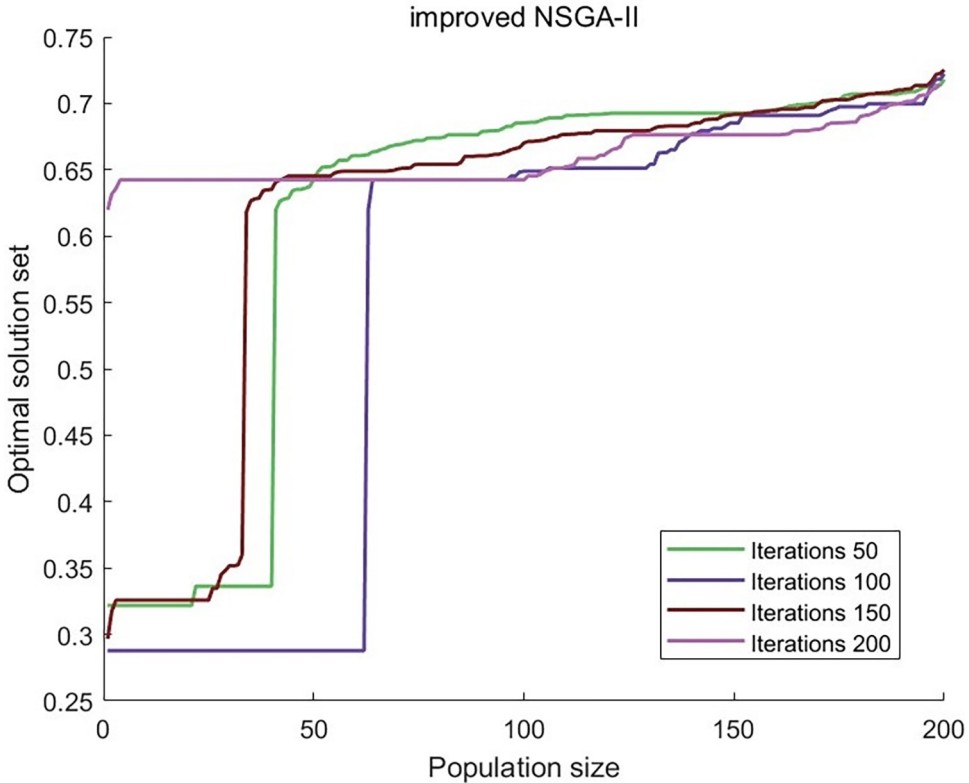

**Fig 12. Using improve NSGA-II to obtain the optimal solution for iterations of 50, 100, 150, and 200.**

## Supporting information

**S1 File.**
(DOCX)

## Author Contributions

**Methodology:** Peng Liu.

**Writing – original draft:** Ying Wang.

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
