## [Decision Letter · Decision Letter 0]

1 Apr 2024

PONE-D-24-05998Shared Manufacturing Service Composition Optimization Based on Bi-Level ProgrammingPLOS ONE

Dear Dr. Liu,

Thank you for submitting your manuscript to PLOS ONE. After careful consideration, we feel that it has merit but does not fully meet PLOS ONE’s publication criteria as it currently stands. Therefore, we invite you to submit a revised version of the manuscript that addresses the points raised during the review process.

We look forward to receiving your revised manuscript.

Kind regards,

Van Thanh Tien Nguyen, Ph.D.

Academic Editor

PLOS ONE

3. We note that your Data Availability Statement is currently as follows: [All relevant data are within the manuscript and its Supporting Information files. The authors declare no conflict of interest. We declare that there is no conflict of interests regarding the publication of this article.]

Reviewers' comments:

Reviewer's Responses to Questions

**Comments to the Author**

1. Is the manuscript technically sound, and do the data support the conclusions?

Reviewer #1: Partly

Reviewer #2: Yes

Reviewer #3: Yes

2. Has the statistical analysis been performed appropriately and rigorously? 

Reviewer #1: I Don't Know

Reviewer #2: Yes

Reviewer #3: Yes

3. Have the authors made all data underlying the findings in their manuscript fully available?

Reviewer #1: No

Reviewer #2: No

Reviewer #3: Yes

4. Is the manuscript presented in an intelligible fashion and written in standard English?

Reviewer #1: Yes

Reviewer #2: No

Reviewer #3: Yes

5. Review Comments to the Author

Reviewer #1: Dear Author(s),

Your manuscript titled “Shared Manufacturing Service Composition Optimization Based on Bi-Level Programming” is now assessed. The topic is really interesting and applicable, but in the current form it needs further processing and obviating shortcomings. I list concerns and issues for your perusal. It can be re-submitted provided the issues are addressed.

1- The English should be improved. The whole proofreading is appreciated.

2- The abstract should be re-written to enhance its quality. Adding the novelty and also numerical achievements in abstract is appreciated. Please put percentage improvement.

3- In abstract, you should mention the objective function is a cost-function which needs to be minimized.

4- What does CRITIC stand for? Please uses extracted from before the first usage of abbreviation. Please perform in all parts.

5- I recommend the title needs changing because it would be more specified when you indicate your algorithm. It seems the current title does not convey the main goal especially NSGA-II.

6- The keywords are insufficient. Please add ample and the most relevant keywords.

7- The used references are imperfect. Please rectify or update all references in terms of data, pages, doi, etc. In addition, more newly published and the most relevant papers should be added in some parts to increase the quality of paper and attracts more researchers’ attention on your manuscript. If possible for you, please omit old-fashioned or un-reputed papers and index-less journals from citation and reference list to enhance the manuscript authentication.

8- One of the most important thing that you used is NSGA-II algorithm. Please explain about it in literature review. Please cite this reference “K. Deb, A. Pratap, S. Agarwal and T. Meyarivan, "A fast and elitist multiobjective genetic algorithm: NSGA-II," in IEEE Transactions on Evolutionary Computation, vol. 6, no. 2, pp. 182-197, April 2002, doi: 10.1109/4235.996017.”

9- In introduction section, please enlist contributions in bullet form, before paper structure is introduced.

10- The re-structure is necessary for your manuscript. It lacks literature review and related work. Please add related work.

11- Related work should be extended. I recommend mention some papers below to improve the quality of current paper. Firstly, introduce NSGA-II, non-dominated solutions, and Pareto set concepts. Please use reference ““K. Deb, A. Pratap, S. Agarwal and T. Meyarivan, "A fast and elitist multiobjective genetic algorithm: NSGA-II," in IEEE Transactions on Evolutionary Computation, vol. 6, no. 2, pp. 182-197, April 2002, doi: 10.1109/4235.996017.”” with suitable citation.

The bi-level programming needs to be introduced with suitable reference. Please add couple of relevant papers in related work.

Service composition technology is used in several industries to reduce overall cost; also, this technology is engaged to improve reliability, security, etc. In IT industry, a decision model paper titled “An iterative mathematical decision model for cloud migration: A cost and security risk approach” was extended to indicate how service composite is beneficial for individuals and businesses.

Bi-objective web service composition algorithm entitled “Bi-objective web service composition problem in multi-cloud environment: a bi-objective time-varying particle swarm optimisation algorithm” was proposed to reduce overall cost and increase security in multi-cloud environment.

Recently published reliability –aware service composition method in paper titled “Reliability-aware web service composition with cost minimization perspective: a multi-objective particle swarm optimization model in multi-cloud scenarios” was proposed to increase system reliability.

A genetic algorithm was extended to solve web service composition problem in “Optimization of Automatic web services composition using genetic algorithm” for distributed systems.

12- Is your algorithm heuristic or meta-heuristic? Why?

13- Since you used GA, what is problem encoding? Termination criteria? Please put condensed sentences about my questions.

14- Is your algorithm scalable?

15- What is implementation environment? Please put condense explanation about implementation/simulation environment. More experiments are recommended to reach concrete results and conclusion.

16- The conclusion should be changed to “Conclusion and future direction”. Please put a condense sentence about limitation of your work. And what is the future direction?

In the current form, I cannot accept it publication, but it can be re-evaluated provided the issues are carefully addressed. Therefore, I recommend Major revision.

Reviewer #2: The authors have addressed my concerns and presented convincing answers to them except my 6th and 7th concerns. I have repeated these concerns below. Please note that based on my 6th concern, you MUST compare your improved algorithm with the conventional one upon implementing both algorithms on a set of popular standard test functions. Furthermore, I requested the authors to revise the English of the manuscript in my 7th comment, however, after reading the revised version of the manuscript; I found out that the English of the paper HAS NOT BEEN IMPROVED enough. The manuscript would be accepted for publication if and only if these two concerns are fully addressed and removed.

6. It is recommended to first test the competence of your improved NSGA-II algorithm on a set of standard benchmark test functions and to compare the results with those of the conventional NSGA-II algorithm prior to go over solving the practical problem introduced in the manuscript.

7. The English of the manuscript can be and should be improved. There are considerable grammatical errors in the text of the manuscript that should be removed in the revised version.

Reviewer #3: As a reviewer, I commend the authors for addressing the critical issue of service composition optimization in shared manufacturing environments. The proposed bi-level programming approach appears to be a promising method for enhancing service quality by considering reliability, trustworthiness, and task fit. Here are my comments and suggestions:

1. Clarity of Abstract:

The abstract is generally well-written, but it could be more concise. It would benefit from a tighter focus on the main contributions of the paper, such as the novel application of the CRITIC method and the improved NSGA-II algorithm.

2. Methodology:

The authors should provide a more detailed explanation of how the CRITIC method is utilized to determine the weights of the indicators. For example, through the analysis of contrast and ambivalence, how to connect to the weighted formula.

3. Case Study and Experimental Results:

It is essential to validate the proposed model with a comprehensive case study or multiple case studies. The experimental results should include a sensitivity analysis to demonstrate how the model responds to variations in input parameters.

4. Future Work:

While the discussion on future work is informative, it would be beneficial to include some direction on how the proposed model can be extended to accommodate dynamic changes in the shared manufacturing environment, such as fluctuating demand or the entry of new service providers.

In summary, this paper presents a valuable addition to the field of shared manufacturing service composition. With further refinement and clarification of the methods and results, it has the potential to make a significant impact on both researchers and practitioners in the field.

6. PLOS authors have the option to publish the peer review history of their article (what does this mean?). If published, this will include your full peer review and any attached files.

Reviewer #1: No

Reviewer #2: No

Reviewer #3: No

---

## [Author Response · Author response to Decision Letter 0]

20 Apr 2024

Dear Editor,

Thank you for allowing a major revision of our manuscript (PONE-D-24-05598) for publication in PLOS ONE, with an opportunity to address the reviewers’ comments.

We are uploading (a) our point-by-point response to the comments, (b) an updated manuscript with yellow highlighting indicating changes, and (c) a clean updated manuscript without highlights.

We thank the referee and the editor very much for the comments and suggestions. We have made considerable effort on revising the paper based on these comments. For future communications, please contact either P. Liu at the addresses listed below.

Yours sincerely,

Peng Liu

Professor Peng Liu 

School of Management

Shenyang University of Technology 

Shenyang 110870

China

E-mail: liupeng@sut.edu.cn

Response to the Comments of Reviewer #1 on PONE-D-24-05998

We thank the referee very much for the comments and suggestions. They are very helpful for us to revise and improve the paper. The paper has been carefully revised according to the referee’s advice. We have made the following changes on the paper accordingly.

Reviewer#1, Concern # 1: The English should be improved. The whole proofreading is appreciated.

Author response: According to referee’s comments, we carefully checked and revised all text writing logic and spelling. The use of English in the revised paper is greatly improved.

We updated the manuscript with the yellow highlighted changes in the revised paper.________________________________________

Reviewer#1, Concern # 2: The abstract should be re-written to enhance its quality. Adding the novelty and also numerical achievements in abstract is appreciated. Please put percentage improvement.

Author response: As suggested by the referee, we have improved the descriptions of the abstract. We have noted in the abstract the percentage improvement in operational efficiency and convergence speed of the improved NSGA-II algorithm compared to traditional algorithm.

We updated the manuscript with the yellow highlighted changes on page 1 in the revised paper.

Reviewer#1, Concern # 3: In abstract, you should mention the objective function is a cost-function which needs to be minimized.

Author response: We thank the referee very much for the comment. In the revised abstract, we have added the upper objective function as the function for minimizing and the lower objective function as the function for maximizing.

We updated the manuscript with the yellow highlighted changes on page 1 in the revised paper.

Reviewer#1, Concern # 4: What does CRITIC stand for? Please uses extracted from before the first usage of abbreviation. Please perform in all parts.

Author response: We thank the referee very much for the comment. The full name of the CRITIC is Criteria Importance Through Intercrieria Correlation. Before using abbreviations, we have indicated the full name of the CRITIC method.

 We updated the manuscript with the yellow highlighted changes on page 1 in the revised paper.

Reviewer#1, Concern # 5: I recommend the title needs changing because it would be more specified when you indicate your algorithm. It seems the current title does not convey the main goal especially NSGA-II.

Author response: We thank the referee very much for the comment. We have changed the title “Shared Manufacturing Service Composition Optimization Based on Bi-Level Programming” to “Bi-Level Optimization of Shared Manufacturing Service Composition Based on Improved NSGA-II”.

We updated the manuscript with the yellow highlighted changes on page 1 in the revised paper.

Reviewer#1, Concern # 6: The keywords are insufficient. Please add ample and the most relevant keywords.

Author response: According to referee’s comments, we have added “Improved NSGA-II algorithm” and “CRITIC method” to the keywords.

We updated the manuscript with the yellow highlighted changes on page 1 in the revised paper.

Reviewer#1, Concern # 7: The used references are imperfect. Please rectify or update all references in terms of data, pages, doi, etc. In addition, more newly published and the most relevant papers should be added in some parts to increase the quality of paper and attracts more researchers’ attention on your manuscript. If possible for you, please omit old-fashioned or un-reputed papers and index-less journals from citation and reference list to enhance the manuscript authentication.

Author response: We thank the referee very much for the comment. We made changes to the references section and omit old-fashioned or un-reputed papers and index-less journals from citation and reference list to enhance the manuscript authentication.

We updated the manuscript with the yellow highlighted changes on pages 28-31 in the revised paper.

Reviewer#1, Concern # 8: One of the most important thing that you used is NSGA-II algorithm. Please explain about it in literature review. Please cite this reference “K. Deb, A. Pratap, S. Agarwal and T. Meyarivan, "A fast and elitist multiobjective genetic algorithm: NSGA-II," in IEEE Transactions on Evolutionary Computation, vol. 6, no. 2, pp. 182-197, April 2002, doi: 10.1109/4235.996017.”

Author response: According to referee’s comments, we cite this reference “K. Deb, A. Pratap, S. Agarwal and T. Meyarivan, A fast and elitist multiobjective genetic algorithm: NSGA-II. IEEE Transactions on Evolutionary Computation, vol. 6, no. 2, pp. 182-197, April 2002, doi: 10.1109/4235.996017.” in Section 2.3.

We updated the manuscript with the yellow highlighted changes on page 4 and page 30 in the revised paper.

Reviewer#1, Concern # 9: In introduction section, please enlist contributions in bullet form, before paper structure is introduced.

Author response: According to referee’s comments, before introducing the structure of the paper, we added the main contributions made by this paper. The contributions of this paper are as follows: ①This paper establishes a bi-level programming model, which is different from previous bi-level programming models. The model targets the supply and demand sides of shared manufacturing, and the lower level indicators include service reliability, service credibility, and task coordination. The service reliability and credibility indicators are decomposed in detail, and the model represents more comprehensive interests. ②This paper uses the Criteria Importance Though Intercrieria Correlation (CRITIC) to determine the weights of indicators and make the upper and lower indicators more fair and objective. ③ This paper has enhanced the genetic algorithm’s selection, crossover, and mutation processes, as well as its efficiency and rate of convergence.

We updated the manuscript with the yellow highlighted changes on page 2 in the revised paper.

Reviewer#1, Concern # 10: The re-structure is necessary for your manuscript. It lacks literature review and related work. Please add related work.

Author response: According to referee’s comments, we have added literature review and related work in Section 2.

We updated the manuscript with the yellow highlighted changes on pages 3-4 in the revised paper.

Reviewer#1, Concern # 11: Related work should be extended. I recommend mention some papers below to improve the quality of current paper. Firstly, introduce NSGA-II, non-dominated solutions, and Pareto set concepts. Please use reference ““K. Deb, A. Pratap, S. Agarwal and T. Meyarivan, "A fast and elitist multiobjective genetic algorithm: NSGA-II," in IEEE Transactions on Evolutionary Computation, vol. 6, no. 2, pp. 182-197, April 2002, doi: 10.1109/4235.996017.”” with suitable citation.

The bi-level programming needs to be introduced with suitable reference. Please add couple of relevant papers in related work.

Service composition technology is used in several industries to reduce overall cost; also, this technology is engaged to improve reliability, security, etc. In IT industry, a decision model paper titled “An iterative mathematical decision model for cloud migration: A cost and security risk approach” was extended to indicate how service composite is beneficial for individuals and businesses.

Bi-objective web service composition algorithm entitled “Bi-objective web service composition problem in multi-cloud environment: a bi-objective time-varying particle swarm optimisation algorithm” was proposed to reduce overall cost and increase security in multi-cloud environment.

Recently published reliability –aware service composition method in paper titled “Reliability-aware web service composition with cost minimization perspective: a multi-objective particle swarm optimization model in multi-cloud scenarios” was proposed to increase system reliability.

A genetic algorithm was extended to solve web service composition problem in “Optimization of Automatic web services composition using genetic algorithm” for distributed systems.

Author response: According to referee’s comments, we have added references and citations to the literature on bi-level programming models in 2.1 and the concepts of NSGA-II, non-dominant ordering, and Pareto solution sets in section 2.3.

Service composition is the main topic of the current research concern. SHIRVANI M H el. [6] presented an iterative mathematical decision model for organizations to evaluate whether to invest in establishing information technology (IT) infrastructure on-premises or outsourcing IT services on a multicloud environment. In order to reduce overall costs and increase security in multi-cloud environment, SHIRVANI M H [7] formulated the web service composition problem as a bi-objective optimization problem with service cost and multi cloud risk perspectives in a ever-increasing multi cloud environment (MCE) in which each provider has its variable pricing strategy and different security level. TABALVANDANI M A N et al. [8] proposed a multi-objective particle swarm optimization model in multi-cloud scenarios to increase system reliability.

We updated the manuscript with the yellow highlighted changes on page 3 and page 28 in the revised paper.

Reviewer#1, Concern # 12: Is your algorithm heuristic or meta-heuristic? Why?

Author response: We thank the referee very much for the comment. The algorithm proposed in this paper is a meta-heuristic. The genetic algorithm itself is a meta-heuristic, which is inspired by the "survival of the fittest" in the process of biological evolution. By randomly generating an initial set of solutions, and then using genetic manipulation to cross and mutate, the quality of the solutions is continuously optimized until the optimal solution is reached. This algorithm is characterized by its versatility and flexibility, and can be applied to a variety of optimization problems. Although we have improved the process of selection, crossover, and mutation, and the improved NSGA-II algorithm has improved performance and efficiency compared with the traditional NSGA-II algorithm, it is still a meta-heuristic.

Reviewer#1, Concern # 13: Since you used GA, what is problem encoding? Termination criteria? Please put condensed sentences about my questions.

Author response: We thank the referee very much for the comment. It is used integer encoding to encode candidate manufacturing resource services to generate chromosome genes corresponding to individuals in the population. For example, for the candidate service composition , the corresponding chromosome encoding method is [1，3，1，2，3]. When the set maximum number of iterations is reached, the algorithm is terminated.

Reviewer#1, Concern # 14: Is your algorithm scalable?

Author response: We thank the referee very much for the comment. The improved NSGA-II algorithm proposed in this paper is scalable. The algorithm's performance growth is flat and able to remain within an acceptable range when dealing with larger or more complex problems. As shown in the table, as the number of iterations increases, the runtime of the algorithm increases at a flat rate.

Iterations Improved run time Traditional run time

100 13.175 44.262

300 38.242 135.591

500 63.836 212.699

Reviewer#1, Concern # 15: What is implementation environment? Please put condense explanation about implementation/simulation environment. More experiments are recommended to reach concrete results and conclusion.

Author response: We thank the referee very much for the comment. This article uses the improved NSGA-II algorithm to solve the model in the MATLAB 2022b computing environment (running on 12th Gen Intel (R) Core (TM) i5-12500H, 2.50 GHz, Windows 11 Home Chinese version, RAM16.0GB), with an initial population NP=200, maximum iteration interval=180, guided crossover probability pc=0.7, and non-uniform mutation probability pm=0.03. These parameters were referenced from earlier literature utilizing genetic algorithms and determined by continual debugging during the experiment.

We updated the manuscript with the yellow highlighted changes on page 22 in the revised paper.

Reviewer#1, Concern # 16: The conclusion should be changed to “Conclusion and future direction”. Please put a condense sentence about limitation of your work. And what is the future direction?

In the current form, I cannot accept it publication, but it can be re-evaluated provided the issues are carefully addressed. Therefore, I recommend Major revision.

Author response: According to referee’s comments, we add limitation of our work and the future direction in the revised paper.

This paper mainly considers the research on the service combination optimization method from the perspective of the supply side and the demand side, but in the actual production and manufacturing process, the supply side of the shared manufacturing platform also has its own demand for the indicators of the whole production process. The next step will be to consider the interests of the three parties from the perspective of the shared manufacturing platform, improve the evaluation index system of the shared manufacturing service composition. For example, it is to incorporate the substitutability of services into the evaluation indicators required by the suppliers of shared manufacturing platforms. It is also to consider introducing a time decay function to handle service evaluation indicators.

We updated the manuscript with the yellow highlighted changes on page 27 in the revised paper.

Response to the Comments of Reviewer #2 on PONE-D-24-05998

We thank the referee very much for the comments and suggestions. They are very helpful for us to revise and improve the paper. The paper has been carefully revised according to the referee’s advice. We have made the following changes on the paper accordingly.

Reviewer#2, Concern # 6: It is recommended to first test the competence of your improved NSGA-II algorithm on a set of standard benchmark test functions and to compare the results with those of the conventional NSGA-II algorithm prior to go over solving the practical problem introduced in the manuscript.

Author response: As suggested by the referee, we select three test functions, DTLZ1, DTLZ2, and DTLZ3, and test the GD and IGD values of the traditional NSGA-II algorithm and improved NSGA-II algorithm for 100, 300, and 500 iterations, respectively. Test each function 10 times and take the average value, and record the algorithm running time for different iterations. The experimental results are shown in Tables 2-3. 

Table 2. GD and IGD values of improved NSGA-II algorithm

Iterations Test the function Improved GD Improved IGD Run time

100 DTLZ1 0.038148 0.059894 

13.175

 DTLZ2 1.091276 0.096658 

 DTLZ3 0.594196 0.

---

## [Decision Letter · Decision Letter 1]

6 May 2024

Bi-Level Optimization of Shared Manufacturing Service Composition Based on Improved NSGA-II

PONE-D-24-05998R1

Dear Dr. Liu,

We’re pleased to inform you that your manuscript has been judged scientifically suitable for publication and will be formally accepted for publication once it meets all outstanding technical requirements.

Kind regards,

Van Thanh Tien Nguyen, Ph.D.

Academic Editor

PLOS ONE

Additional Editor Comments (optional):

Reviewers' comments:

Reviewer's Responses to Questions

**Comments to the Author**

1. If the authors have adequately addressed your comments raised in a previous round of review and you feel that this manuscript is now acceptable for publication, you may indicate that here to bypass the “Comments to the Author” section, enter your conflict of interest statement in the “Confidential to Editor” section, and submit your "Accept" recommendation.

Reviewer #1: (No Response)

Reviewer #2: All comments have been addressed

2. Is the manuscript technically sound, and do the data support the conclusions?

Reviewer #1: (No Response)

Reviewer #2: Yes

3. Has the statistical analysis been performed appropriately and rigorously? 

Reviewer #1: (No Response)

Reviewer #2: Yes

4. Have the authors made all data underlying the findings in their manuscript fully available?

Reviewer #1: (No Response)

Reviewer #2: Yes

5. Is the manuscript presented in an intelligible fashion and written in standard English?

Reviewer #1: (No Response)

Reviewer #2: Yes

6. Review Comments to the Author

Reviewer #1: Dear Author(s),

Thanks for your attempt to obviate shortcomings,

The issues have been addressed. The manuscript is eligible for publication.

Reviewer #2: I appreciate the efforts of the authors to improve the technical aspects of the manuscript. The manuscript is accepted for publication as is.

7. PLOS authors have the option to publish the peer review history of their article (what does this mean?). If published, this will include your full peer review and any attached files.

Reviewer #1: No

Reviewer #2: No

---

## [Editor Report · Acceptance letter]

13 May 2024

PONE-D-24-05998R1 

PLOS ONE

Dear Dr. Liu, 

I'm pleased to inform you that your manuscript has been deemed suitable for publication in PLOS ONE. Congratulations! Your manuscript is now being handed over to our production team.

Kind regards, 

on behalf of

Asst. Prof. Van Thanh Tien Nguyen 

Academic Editor

PLOS ONE